# Dysregulation of mitochondrial dynamics proteins are a targetable feature of human tumors

Gray R. Anderson[1], Suzanne E. Wardell[1], Merve Cakir[2], Catherine Yip[1], Yeong-ran Ahn[1], Moiez Ali[1], Alexander P. Yllanes[1], Christina A. Chao[1], Donald P. McDonnell[1] & Kris C. Wood[1]

Altered mitochondrial dynamics can broadly impact tumor cell physiology. Using genetic and pharmacological profiling of cancer cell lines and human tumors, we here establish that perturbations to the mitochondrial dynamics network also result in specific therapeutic vulnerabilities. In particular, through distinct mechanisms, tumors with increased mitochondrial fragmentation or connectivity are hypersensitive to SMAC mimetics, a class of compounds that induce apoptosis through inhibition of IAPs and for which robust sensitivity biomarkers remain to be identified. Further, because driver oncogenes exert dominant control over mitochondrial dynamics, oncogene-targeted therapies can be used to sensitize tumors to SMAC mimetics via their effects on fission/fusion dynamics. Collectively, these data demonstrate that perturbations to the mitochondrial dynamics network induce targetable vulnerabilities across diverse human tumors and, more broadly, suggest that the altered structures, activities, and trafficking of cellular organelles may facilitate additional cancer therapeutic opportunities.

[1] Department of Pharmacology and Cancer Biology, Duke University, Durham, NC 27710, USA. [2] Program in Computational Biology and Bioinformatics, Duke University, Durham, NC 27710, USA. Correspondence and requests for materials should be addressed to K.C.W. (email: kris.wood@duke.edu)

Mitochondria exist along a dynamic continuum between fragmented and fused states. Fission/fusion dynamics regulate mitochondrial metabolism and apoptosis, and emerging data suggest that tumors alter mitochondrial dynamics homeostasis to promote their growth and survival[1–6]. Previous studies have demonstrated that mitochondrial dynamics regulate critical cellular, physiological, and pathophysiological processes that include apoptosis, cellular metabolic programs, and mitochondrial health[7–10]. For example, sumoylation of a critical fission protein, dynamin-related protein 1 (Drp1), is required to maintain the endoplasmic reticulum (ER)-mitochondrial signaling network necessary for apoptosis[11]. Further, recent work has established that mitochondrial fission is important for the normal physiological clearance of apoptotic cells by macrophages[12]. Alterations in mitochondrial dynamics are also implicated in various disease states, including diabetic stress (i.e., high-glucose-induced ROS) and neurodegeneration, the latter of which is associated with disruption of fission/fusion cycles[13,14]. Lastly, dysregulation of mitochondrial dynamics is a key feature of aging; for example, loss of optic atrophy 1 (OPA1), a key mitochondrial fusion protein, contributes to skeletal muscle loss in aging mice[7,15].

Because mitochondrial dynamics broadly impact cellular apoptosis and metabolism, it is perhaps unsurprising that emerging studies have begun to demonstrate that tumors alter their mitochondrial dynamics homeostasis to promote their growth and survival[16]. Notably, signaling downstream of mutant KRAS in pancreatic cancers leads to mitochondrial fragmentation and increased activation of Drp1, processes that are required for KRAS-driven tumor growth in vivo[3,4]. In addition, recent studies also suggest that mitochondrial dynamics are important for regulating metastatic phenotypes such as invasion and migration in breast and thyroid cancers[17,18].

In light of the observation that mitochondrial dynamics are frequently altered in human cancers and the likelihood that these alterations broadly impact cell physiology, there exists an imperative to define therapeutic vulnerabilities driven by changes in mitochondrial dynamics networks. If identified, such vulnerabilities could have a substantial impact in cancers for which the primary oncogenic driver is either unknown or undruggable (e.g., pancreatic ductal adenocarcinoma (PDAC), high-grade serous ovarian cancer (HGSOC), and triple-negative breast cancer (TNBC)). Further, even in settings like BRAF mutant melanoma and EGFR mutant non-small-cell lung cancer (NSCLC), where effective precision therapies have been established, targeted therapies often yield incomplete and transient responses[19–22]. Thus, the discovery of vulnerabilities associated with altered mitochondrial dynamics could lead to both new targeted therapies for tumors that have been historically refractory to such approaches as well as strategies to augment the activity of existing drugs. In this study, we use genomic and pharmacological approaches to establish that perturbations to mitochondrial dynamics regulating proteins lead to targetable vulnerabilities.

## Results

### Computational proof-of-principle for altered drug sensitivity.
To explore the hypothesis that dysregulated mitochondrial dynamics may impact drug sensitivity, we began by examining the alteration status of six canonical dynamics-regulating genes - OPA1, MFN1, MFN2, FIS1, MFF, and DNM1L (which encodes Drp1)—in human cancers. Using publically available datasets from The Cancer Genome Atlas (TCGA), we found that four of these genes—OPA1, MFN1, FIS1, and DNM1L—are recurrently amplified in certain cancers at frequencies exceeding 5%: HGSOC, breast cancer, head and neck squamous cell carcinoma

(HNSCC), pancreatic cancer, and lung cancer (using a pan-analysis of lung adenocarcinomas and lung squamous cell carcinomas) (Fig. 1a)[19,20,23–26]. By contrast, we observed no evidence of a recurrent copy number loss or mutations in these genes. To examine whether tumors with amplifications in mitochondrial dynamics-regulating genes harbor unique drug sensitivities, we queried publically available drug sensitivity data from over 1000 genomically annotated cancer cell lines treated with a panel of 265 anti-cancer drugs[27]. Specifically, we classified cell lines from each of the tissues above as either amplified or non-amplified for each of these genes, controlling for amplifications in neighboring oncogenes KRAS (DNM1L), BRAF (FIS1), mTOR (MFN2), and PIK3CA (MFN1 and OPA1). Next, we searched for drugs with differential potency in amplified vs. non-amplified cell lines using a Benjamini–Hochberg corrected p-value threshold of $p < 0.05$ by Welch's t-test. In many cases, differences in sensitivity to distinct classes of drugs were evident when cell lines were stratified based on the amplification status of dynamics-regulating genes (Fig. 1b, Supplementary Data 1). For example, breast cancer cell lines harboring DNM1L amplifications exhibited hypersensitivity to the ER stress-inducing drug thapsigargin and the XIAP inhibitor embelin as well as resistance to chemically distinct inhibitors of the phosphoinositide 3-kinase (PI3K) pathway (Fig. 1c, d). Further, preliminary analysis of breast cancer and melanoma cell line models suggests that amplifications in mitochondrial dynamics genes can affect mitochondrial morphology (Supplementary Fig. 1a, b). By contrast, a similar analysis in pancreatic cancer cell lines failed to detect changes in mitochondrial morphology in a cell line model with amplified OPA1, potentially because of the well-established impact of mutant KRAS on Drp1 activation (Supplementary Fig. 1c)[3,4]. Collectively, these data demonstrate that canonical mitochondrial dynamics-regulating genes are recurrently amplified in human cancers and suggest that cancers with amplifications in dynamics regulating genes may harbor unique and actionable drug sensitivities.

### Isogenic screening identifies drug with differential sensitivity.
To broadly define the impact of altered mitochondrial dynamics proteins on drug sensitivity, we next used high-throughput drug screening in isogenic cell lines. Specifically, we first generated isogenic models of dysregulated dynamics in a HGSOC cell line (TYKNU) using CRISPR-Cas9-mediated knockout of OPA1 or DNM1L and confirmed that these manipulations increased mitochondrial fragmentation or connectivity, respectively (Fig. 2a, Supplementary Fig. 2a). Isogenic derivatives were then screened in duplicate with a ~2100 compound drug library at two different drug doses (2 and 10 μM) (Fig. 2b, Supplementary Data 2). After 72 h, cell viability was determined, then normalized to vehicle treatment to account for differences in growth rates. Hits were determined by calculating the ratio of normalized viabilities for each drug in knockout derivatives compared to cells expressing a non-targeted control sgRNA. Drugs with increased potency in knockout cells were identified as those scoring ~2.5 standard deviations away from the mean ($log_2$(knockout/control) $< -0.5$), whereas drugs with decreased potency in knockout cells were identified as those scoring ~3.5 standard deviations from the mean ($log_2$(knockout/control) $> 0.7$), where in both cases stringent scoring thresholds were chosen to exceed the full range of scores observed in 816 empty control wells. (Fig. 2b, Supplementary Fig. 2b, c).

Dysregulated mitochondrial dynamics (through OPA1 or DNM1L loss), both in the direction of increased fragmentation and increased connectivity, altered drug sensitivities, conferring resistance to some drugs and sensitivity to others (Fig. 2c, Supplementary Fig. 2d). Specifically, increasing mitochondrial

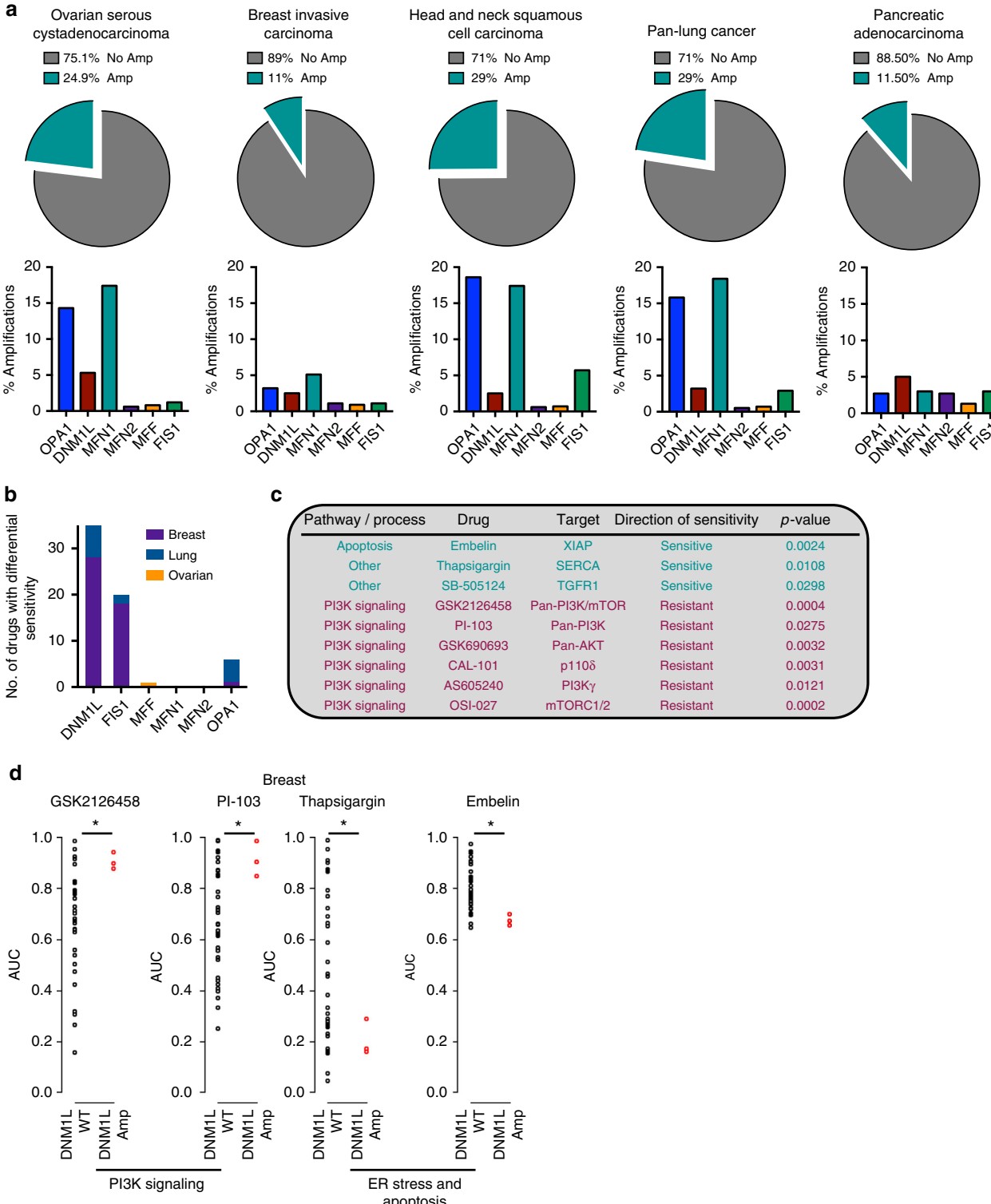

**Fig. 1** Computational proof-of-principle that cancers with altered mitochondrial dynamics exhibit differential drug sensitivity. **a** Top, pie charts in five major cancer types summarizing the percentage of tumors with amplifications in *DNM1L*, *OPA1*, *MFF*, *MFN1*, *MFN2*, or *FIS1*, using publicly available data from the TCGA. Bottom, bar graphs summarizing the percentage of amplifications in each of the six genes from above. **b** The number of drugs from a dataset of 265 drugs with statistically significant differences in sensitivity when cell lines of a given cancer type are stratified based upon amplification status of each of the six genes listed above making sure to control for amplifications in neighboring oncogenes. **c** Examples of the drug sensitivity differences from **b** in *DNM1L* amplified breast cancer. **d** Four drugs from **c** are shown to illustrate the differences in drug sensitivity between *DNM1L*-amplified breast cancer vs. wild-type breast cancer. *$p < 0.05$ by Welch's *t*-test. AUC area under the curve for the drug sensitivity assay

fragmentation via *OPA1* loss tended to cause increased resistance to drug treatments—consistent with published studies which, when taken together, indicate that a fragmented mitochondrial network is associated with general chemo-resistance[28–30]—while increasing connectivity via *DNM1L* loss tended to cause increased drug sensitivity (Fig. 2d). We observed classes of drugs with increased potency in cells with increased fragmentation, increased connectivity, or both (Supplementary Fig. 2d), a diverse selection of which were individually validated in secondary, eight-point growth inhibition-50% (GI$_{50}$) assays (Supplementary Fig. 2e). For example, while sensitivity to drugs targeting autophagic processes were unaffected by alterations in mitochondrial dynamics, drugs

targeting transmembrane transporters often exhibited increased potency in cells with increased connectivity (but not increased fragmentation), and a subset of drugs targeting apoptotic processes exhibited increased potency in cells exhibiting increased fragmentation or increased connectivity (Fig. 2e). A particularly interesting example of the latter phenomenon were drugs targeting nucleotide metabolism (Fig. 2f). Within this class, drugs targeting purine metabolism showed enhanced potency in cells with increased fragmentation or connectivity, while drugs targeting pyrimidine metabolism did not. Further, sub-stratification of purine targeting drugs based on the nucleotide targeted by each drug (adenine or guanine) revealed that guanine

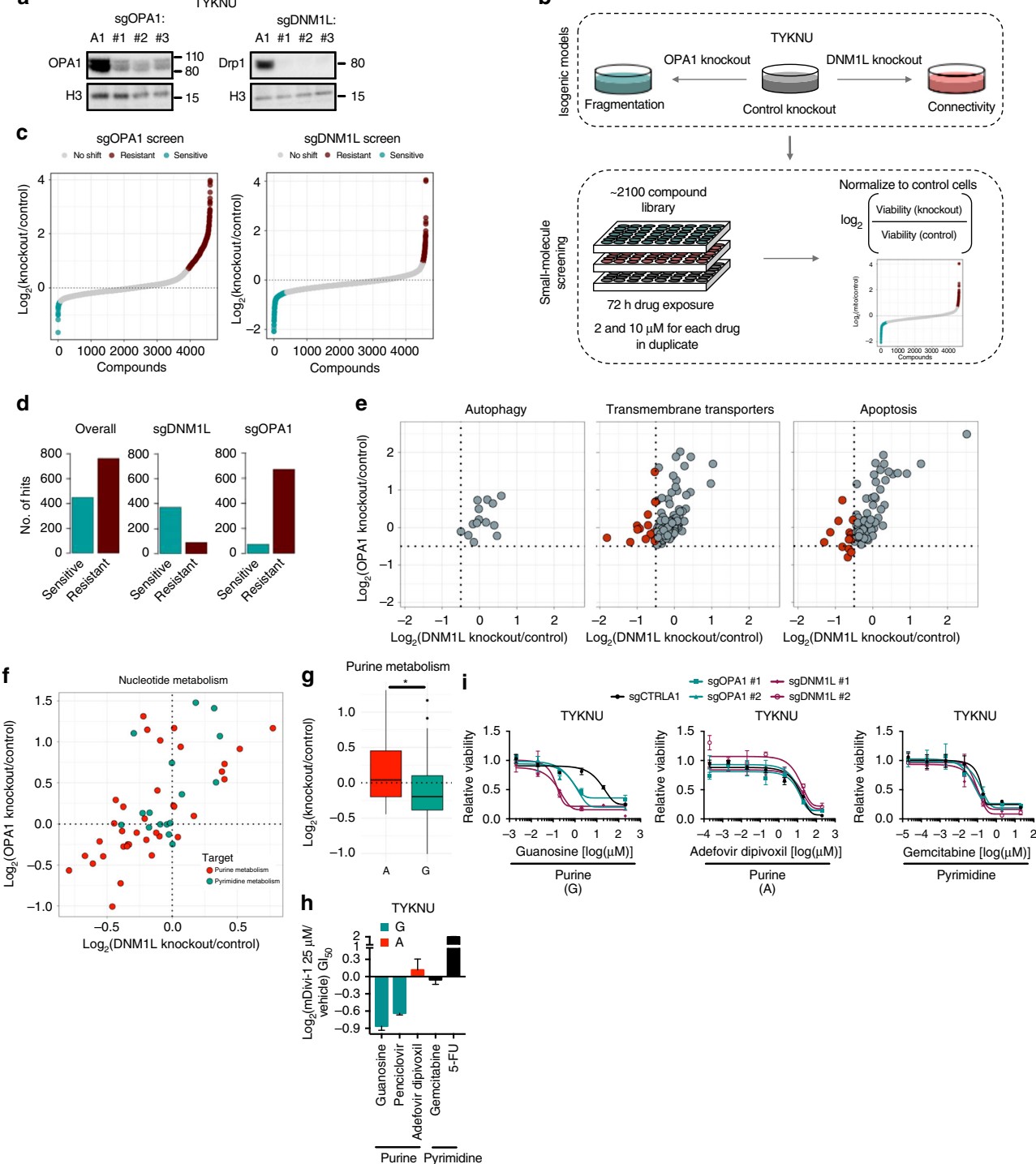

targeting drugs exhibited increased potency in both isogenic derivatives, while adenine targeting drugs exhibited no change in potency (Fig. 2g). The increased potency of agents targeting guanine metabolism in cells with altered dynamics was validated in secondary $GI_{50}$ assays using drugs targeting guanine (guanosine, penciclovir) vs. adenine (adefovir, dipivoxil) or pyrimidine metabolism (Gemcitabine and 5-FU), using both genetic approaches and pharmacological Drp1 inhibition with the compound mDivi-1 (Fig. 2h, i). Collectively, these data suggest that loss of OPA1 or DNM1L can impact drug sensitivity, and that certain classes of drugs exhibit increased potency in cells with increased fragmentation or increased connectivity.

**Small-molecule screens nominate SMAC mimetics as a vulnerability.** As a class, drugs targeting apoptosis contained the highest fraction of agents with increased potency in cells with both increased fragmentation and increased connectivity. Within this category, we noted a particularly striking enrichment for SMAC mimetics—inhibitors of the pro-survival inhibitor of apoptosis (IAP) family of proteins that exhibit low potency as single agents in most solid tumors. Specifically, all five SMAC mimetics within the library of ~2100 screened compounds scored with increased potency in cells with altered dynamics, often in both OPA1 and DNM1L knockout derivatives, implying that cells with altered levels of dynamics regulating proteins exhibit an increased dependence on IAP proteins (Fig. 3a). We validated these findings using genetic knockouts of OPA1 and DNM1L in eight-point $GI_{50}$ assays (Fig. 3b). In addition, we validated that the activities SMAC mimetics could be phenocopied using genetic knockdown of XIAP, CIAP1, and CIAP2 in standard growth assays (Supplementary Fig. 3a). Further, the relationship between altered dynamics protein levels and sensitivity to SMAC mimetics was not a unique feature of TYKNU cells or HGSOCs, as knockout of OPA1 or DNM1L similarly potentiated the toxicity of SMAC mimetics in cell lines derived from non-high-grade ovarian carcinoma (MCAS and TOV-112), lung adenocarcinoma (PC9), melanoma (A375), breast cancer (T47D), and pancreatic cancer (Panc03.27) (Fig. 3c–f and Supplementary Fig. 3b, c).

Translationally, these findings could have important implications. First, they suggest that tumors with genomic amplification of canonical mitochondrial dynamics-regulating genes (OPA1, DNM1L, MFN1, MFN2, FIS1, and MFF) may have increased sensitivity to SMAC mimetics in the event that such amplifications engender structural changes to the mitochondrial network—a relationship that has not been fully elucidated. Nonetheless, in panels of breast and lung cancer cell lines, we observed increased, often submicromolar sensitivity to SMAC mimetics in cell lines with copy number amplifications, while cell lines lacking amplifications were insensitive (Fig. 3g, h). Second, these findings

suggest that direct pharmacological dysregulation of mitochondrial dynamics in tumors lacking genomic amplifications in dynamics-regulating genes may potentiate the toxicity of SMAC mimetics. Consistent with this hypothesis, we observed that Drp1 inhibition with mDivi-1 potentiated the toxicity of SMAC mimetics in ovarian, lung, breast, and melanoma cell lines (Supplementary Fig. 3d–f). These results were further validated in an orthotopic, in vivo model of advanced endocrine therapy resistant breast cancer, where co-treatment with mDivi-1 and the SMAC mimetic BV6 reduced tumor growth and extended survival relative to single agent treatments, without evidence of toxicity (Fig. 3i, Supplementary Fig. 3g). It is worth noting that mDivi-1 has been shown to have significant off-target effects related to inhibition of electron transport chain (ETC) function[31]. However, the fact that mDivi-1 treatment phenocopies the effect of DMN1L knockout, combined with the observation that mDivi-1 activity is abrogated in DMN1L knockout cells, suggests that the observed pharmacological sensitization is occurring via the on-target inhibition of Drp-1 (Supplementary Fig. 3h). Further, these findings may have broader implications, as they suggest a general reliance on anti-apoptotic function to prevent death in the context of OPA1 or DNM1L knockout, offering additional targets for therapy. Indeed, we observed that OPA1 and DNM1L knockout cells also exhibit increased susceptibility to compounds such as the dual BCL-2/BCL-$X_L$ inhibitor ABT737 and the MCL-1 inhibitor S63845, but not to general apoptosis-inducing compounds such as etoposide (Supplementary Fig. 3i). Together, these data demonstrate that diverse tumors with genetically or pharmacologically induced perturbations to mitochondrial dynamics regulating proteins are vulnerable to treatment with SMAC mimetics and other targeted apoptosis-inducing compounds.

**Drp1 loss leads to apoptosis through leakiness of cytochrome c.** Having established that tumors with dysregulated mitochondrial dynamics are hypersensitive to SMAC mimetics, we turned our attention to understanding the mechanisms that regulate drug sensitivity in cells with increased connectivity. First, using several structurally distinct SMAC mimetics we observed, in multiple cell lines, that sensitivity was associated with increased apoptosis, as combined genetic or pharmacological inhibition of IAPs and Drp1 led to increases in annexin V+ staining (Fig. 4a, Supplementary Fig. 4a–d). Further, apoptosis induction could be fully rescued via treatment with the pan-caspase inhibitor Q-VD-OPh (Supplementary Fig. 4b). IAP proteins inhibit apoptosis downstream of cytochrome c release from depolarized mitochondria, and a recent report demonstrated in colorectal cancer cells that long-term suppression of Drp1 led to leakiness of cytochrome c[32]. Consistent with this concept, subcellular fractionation experiments in MCAS and A375 cells transduced with Cas9 and either a

**Fig. 2** 2100 compound small-molecule screening in isogenic models of dysregulated dynamics uncovers differential drug sensitivities. **a** Immunoblot of OPA1, Drp1, or a loading control in TYKNU cells transduced with a control CRISPR or three independent sgRNAs targeting either OPA1 or DNM1L. Immunoblots are cropped for clarity. Immunoblot is representative of three independent experiments. **b** Schematic of the small-molecule screening method. **c** Left, snake plot depicting the $log_2$(sgOPA1/sgCTRLA1) cellular viability ratio across 2100 compounds at two doses per compound. Right, snake plot depicting the $log_2$(sgDNM1L/sgCTRLA1) cellular viability ratio across 2100 compounds at two doses per compound. Blue indicates a sensitivity, red indicates a resistance. **d** A count of the number of hits in each of the conditions listed. **e** A plot showing the $log_2$(sgOPA1/sgCTRLA1) vs. the $log_2$(sgDNM1L/sgCTRLA1) cellular viability ratios across three classes of drugs involving autophagy, transmembrane transporters, and apoptosis. Red indicates a compound that passes the hit-calling threshold for at least one knockout derivative. **f** A plot showing the $log_2$(sgOPA1/sgCTRLA1) vs. the $log_2$(sgDNM1L/sgCTRLA1) cellular viability ratios for drugs targeting nucleotide metabolism. Red indicates drugs that target purine metabolism and blue indicates drugs that target pyrimidine metabolism. **g** A boxplot showing the $log_2$(knockout/control) cellular viability ratio for drugs that target guanine vs. those that target adenine. Data are mean ± SEM. *$p < 0.05$ by one-way ANOVA. **h** The $log_2$(mDivi-1 25 μM/vehicle) $GI_{50}$ for purine targeting drugs guanosine, penciclovir, adefovir dipivoxil, or pyrimidine targeting drugs, gemcitabine or 5-FU. Data are mean ± SEM. *$p < 0.05$ by one-way ANOVA. **i** $GI_{50}$ curves for guanosine, adefovir dipivoxil, and gemcitabine in TYKNU cells transduced with a control targeting sgRNA or three independent sgRNAs targeting either OPA1 or DNM1L ($n = 3$). Data are mean ± SEM. *$p < 0.05$ by one-way ANOVA

non-targeting sgRNA or a *DNM1L* targeting sgRNA revealed increased cytosolic levels of cytochrome c in cells with increased mitochondrial connectivity (Fig. 4b, Supplementary Fig. 4e).

After confirming cytochrome c leakage into the cytosol in cells lacking *DNM1L*, we reasoned that this should result in low levels of caspase 9 cleavage, since this is the initiator caspase that

interacts with apaf-1 to form the apoptosome downstream of mitochondrial cytochrome c release. By contrast, cytochrome c leakage should not cause caspase 8 cleavage, as this is the initiator caspase primarily involved in extrinsic apoptosis[33]. Indeed, immunoblotting of c-caspase 9 in MCAS cells lacking *DNM1L* revealed low levels of steady-state caspase 9 cleavage, a result that

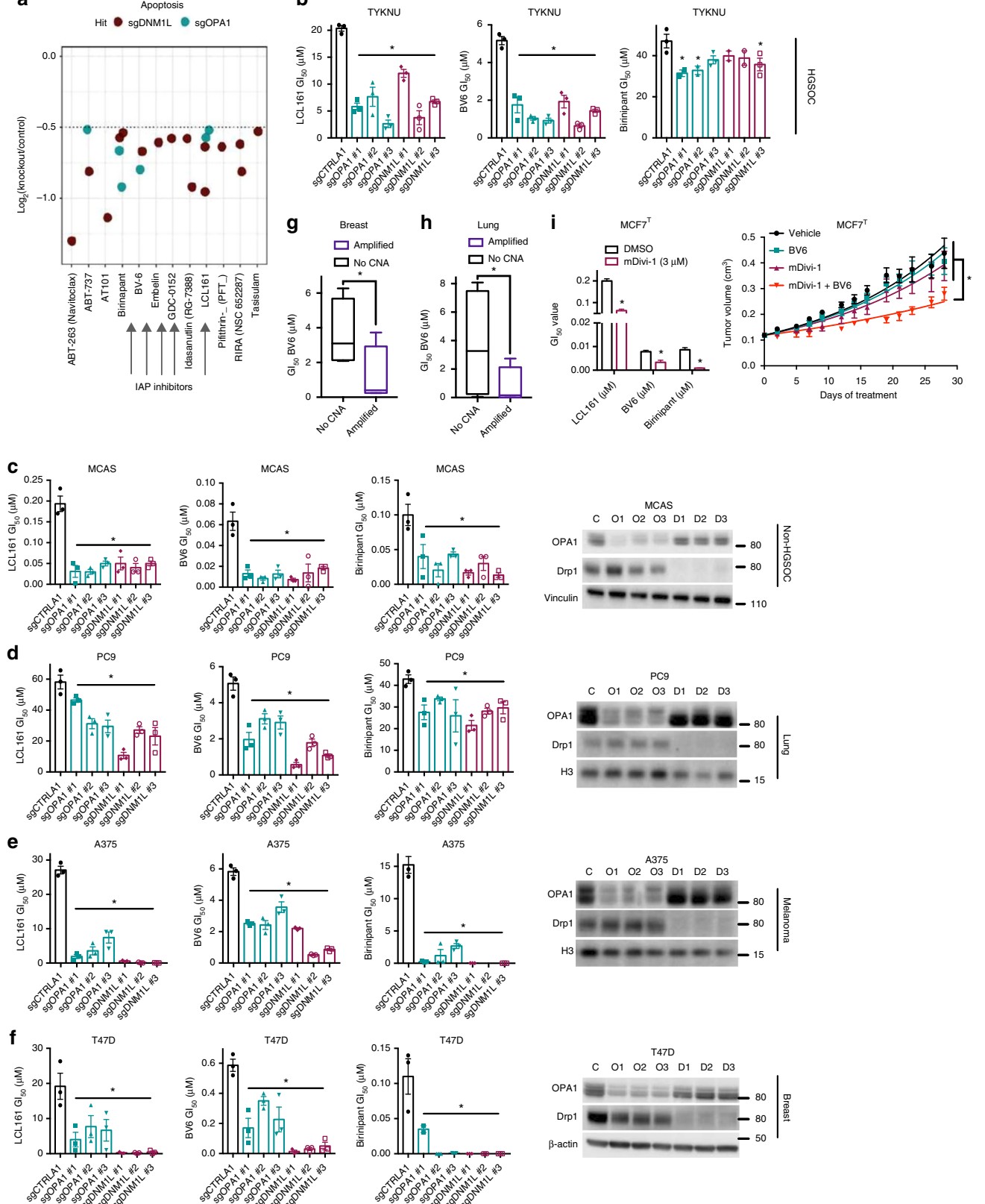

was independently validated in a secondary luminescence assay (Fig. 4c), whereas we were unable to detect c-caspase 8 cleavage through immunoblotting or a secondary luminescence assay (Supplementary Fig. 4f). Finally, to understand the nature of the changes occurring in the function and integrity of the mitochondria that facilitate passive release of cytochrome c, we assessed changes in mitochondrial membrane potential and steady-state levels of ROS. Consistent with recent evidence, we found that *DNM1L* loss led to decreased mitochondrial membrane potential and increased levels of ROS (mitochondrial superoxide and nitric oxide) (Fig. 4d, e)[28]. These changes were functionally important for SMAC mimetic sensitivity, as treatment with the non-specific ROS scavenger *N*-acetyl-cysteine (NaC) and the superoxide specific scavenger, Tiron, rescued the sensitivity of *DNM1L* knockout cells to BV6, while the nitric oxide specific scavenger exhibited weaker effects, suggesting that ROS induction is important for SMAC mimetic mediated cell death and likely occurs upstream of cytochrome c release (Fig. 4f). Lastly, our data suggest a model whereby most cells undergo incomplete mitochondrial outer membrane permeabilization (iMOMP). Consistent with this, immunofluorescence revealed that cells lacking *DNM1L* had, on average, lower co-localization of cytochrome c and the mitochondrial network, as evidenced by a lower Pearson correlation (Supplementary Fig. 4g, h), whereas we did not observe changes in the number of cells with complete MOMP. Together, these results suggest that increasing mitochondrial connectivity via genetic loss of *DNM1L* or pharmacological Drp1 inhibition leads to increased ROS levels and decreased membrane potential, resulting in the leakage of cytochrome c, low level, steady-state caspase 9 cleavage, and potentiation of apoptosis induction by SMAC mimetics.

**OPA1 loss leads to apoptosis through induction of eIF2α-ATF4-CHOP.** After establishing the mechanism(s) underlying sensitivity to SMAC mimetics in cells with increased mitochondrial connectivity, we hypothesized that similar changes may be occurring in mitochondrial function and integrity in cells with increased fragmentation. Indeed, we observed increases in annexin V+ cells, decreased membrane potential, and increased steady-state ROS levels (mitochondrial superoxide and nitric oxide) in cells lacking *OPA1* relative to controls (Fig. 5a–c). However, unlike cells with increased connectivity, cells with increased fragmentation did not show evidence of cytochrome c leakage into the cytoplasm, as demonstrated through subcellular fractionation experiments in MCAS and A375 cells transduced with Cas9 and either a non-targeting sgRNA or a *OPA1* targeting sgRNA (Supplementary Fig. 5a, b).

To understand the molecular events that govern sensitivity to SMAC mimetics in cells with increased mitochondrial fragmentation, we considered recent reports demonstrating that loss of *OPA1* in various cellular contexts can induce the unfolded protein response (UPR) and influence apoptosis regulation[7,34–36]. Immunoblotting of key proteins involved in the UPR and apoptosis signaling revealed increased activation of the eIF2α–ATF4–CHOP axis, along with elevated baseline levels of cleaved caspases and PARP, in cells lacking *OPA1* (Fig. 5d, Supplementary Fig. 5k). Importantly, ROS induction, UPR activation, and caspase cleavage were functionally consequential, as NaC (general ROS scavenger), Tiron (superoxide scavenger), Carboxy-PTIO (nitric oxide scavenger), tauroursodeoxycholic acid (Tudca, a chemical chaperone), GSK2606414 (a PERK inhibitor), and Q-VD-OPh treatment each reversed the sensitivity of *OPA1* knockout cells to BV6 (Fig. 5e–g). To define the epistatic nature of these events, we analyzed ATF4 induction and cleaved effector caspase-7 levels in *OPA1* knockout cells following treatment with the above compounds (Fig. 5h). As expected, Q-VD-OPh rescued caspase cleavage but not ATF4 induction, placing caspase cleavage as a terminal event. Tudca rescued ATF4 induction—consistent with interference with UPR activation—but also rescued caspase cleavage, indicating that induction of the UPR lies upstream of caspase cleavage. Lastly, NaC rescued both ATF4 and caspase cleavage, indicating that ROS induction is upstream of both UPR induction and caspase cleavage (Fig. 5h). Together, our data suggest a model wherein increased mitochondrial fragmentation and connectivity lead to increased steady-state levels of ROS and mitochondrial membrane depolarization. In cells with increased fragmentation, these events lead to induction of the UPR, while in cells with increased connectivity, these events lead to cytosolic leakage of cytochrome c and consequent increases in cleavage of initiator caspase 9. These mechanisms converge downstream to drive increased steady-state levels of effector caspases, rendering cells reliant on IAP proteins. Thus, cells with perturbed mitochondrial dynamics are vulnerable to pharmacological inhibition of IAPs. This model is summarized in Fig. 5i.

**Targeted therapies sensitize to SMAC mimetics by altering dynamics.** Interestingly, recent work has demonstrated that oncogenes exert dominant control over mitochondrial dynamics homeostasis, and that their ability to regulate mitochondrial dynamics is essential for their transforming activities[3,4,37,38]. Based on these data, we reasoned that oncogene targeted therapies should disrupt fission/fusion dynamics in oncogene-driven cancers, sensitizing these cells to treatment with SMAC mimetics. To test this hypothesis, we first confirmed that treatment of three oncogene-driven cancer models (*EGFR* mutant NSCLC, *BRAF* mutant melanoma, and *KRAS* mutant PDAC) with their cognate inhibitors (the EGFR inhibitor gefitinib, the BRAF inhibitor PLX4720, and the ERK inhibitor

**Fig. 3** Dysregulated mitochondrial dynamics confer sensitivity to SMAC mimetics in many cancer types. **a** Log$_2$(knockout/control) cellular viability ratios for compounds that score as sensitive hits in the apoptosis category. Arrows point to compounds that are IAP inhibitors. **b** GI$_{50}$ values for three SMAC mimetics (LCL161, BV6, Birinipant) in TYKNU cells transduced with a control targeting sgRNA or three independent sgRNAs targeting either *OPA1* or *DNM1L* ($n = 3$). Data are mean ± SEM. *$p < 0.05$ by one-way ANOVA. **c** Same as **b** but using MCAS cells. Right, immunoblot of OPA1 or Drp1 in MCAS cells transduced with a control targeting sgRNA, or three independent sgRNAs targeting either *OPA1* or *DNM1L*. Immunoblots are representative of three independent experiments. **d** Same as **c** but using PC9 cells. **e** Same as **c** but using A375 cells. **f** Same as **c** but using T47D cells. **g** GI$_{50}$ value for BV6 in breast cancer cell lines stratified as either amplified in one of six genes (*DNM1L, OPA1, MFF, MFN1, MFN2, FIS1*) (BT20, MDAMB453, BT474, and MCF10A) or wild-type for all six genes (HCC1395, T47D, MDAMB436, CAL51) ($n = 3$). Data are mean ± SEM. *$p < 0.05$ by Student's *t*-test. **h** GI$_{50}$ value to BV6 in lung cancer cell lines stratified as either amplified in one of six genes (*DNM1L, OPA1, MFF, MFN1, MFN2, FIS1*) (CALU-6, H1703, CALU-1, H2228) or wild-type for all six genes (H1048, NCIH596, A549, HCC827) ($n = 3$). Data are mean ± SEM. *$p < 0.05$ by Student's *t*-test. **i** GI$_{50}$ values for three SMAC mimetics (LCL161, BV6, Birinipant) in an advanced model of endocrine-resistant breast cancer, MCF7$^T$ ($n = 3$), treated with either vehicle or mDivi-1 (3 μM). Data are mean ± SEM. *$p < 0.05$ by Student's *t*-test. Orthotopic xenograft model showing tumor volume over time in cohorts treated with either vehicle, mDivi-1 (Drp1 inhibitor), MLN-0128 (mTORC1/2 inhibitor), or the combination of both (see Methods for dosing, $n$ in each group, and statistics)

VX-11e, respectively) resulted in changes to the mitochondrial network morphology (Fig. 6a, b, Supplementary Fig. 6a). Treatment of oncogene-driven cancer models with their cognate targeted therapies also sensitized these models to SMAC mimetics, phenocopying the effects observed following direct perturbation of mitochondrial dynamics (Fig. 6c, Supplementary Fig. 6b). Importantly, oncogene targeted therapies sensitized cancer cells to SMAC mimetics via their effects on mitochondrial dynamics, as genetic inhibition of targeted therapy-induced changes to the mitochondrial network reversed sensitivity to SMAC mimetics (Fig. 6d, Supplementary Fig. 6c, d). Finally, the ability of targeted therapies to potentiate the toxicity of SMAC mimetics can be leveraged to block tumor growth in vivo. Specifically, in an orthoptopic model of advanced endocrine therapy resistant, PIK3CA mutant breast cancer, we observed that combined, low-dose treatment with the mTORC1/2 inhibitor MLN-0128 and the SMAC mimetic BV6 blocked tumor growth and extended survival without evidence of substantial toxicity (Fig. 6e, Supplementary Fig. 6e, f).

## Discussion

In this study, by integrating data from hundreds of cancer cell lines and thousands of human tumors, we discovered that genes involved in the canonical regulation of mitochondrial dynamics are frequently amplified in human cancers, and that these alterations engender in cells drug vulnerabilities. Specifically, a large-scale chemical screen performed in isogenic cell lines revealed that alterations in mitochondrial dynamics regulating proteins directly modulate sensitivity to diverse classes of drugs. Included among these are SMAC mimetics, activators of the mitochondrial apoptosis pathway that have been the subject of substantial translational interest but to date have failed as single agents in unselected patients, in part because they have yet to be associated with robust sensitivity biomarkers[39]. The finding that tumors with genomic amplifications in dynamics-regulating genes, or those in which dynamics have been altered through pharmacological Drp1 inhibition, are hypersensitive to treatment with SMAC mimetics is particularly notable given the field's increasing appreciation of

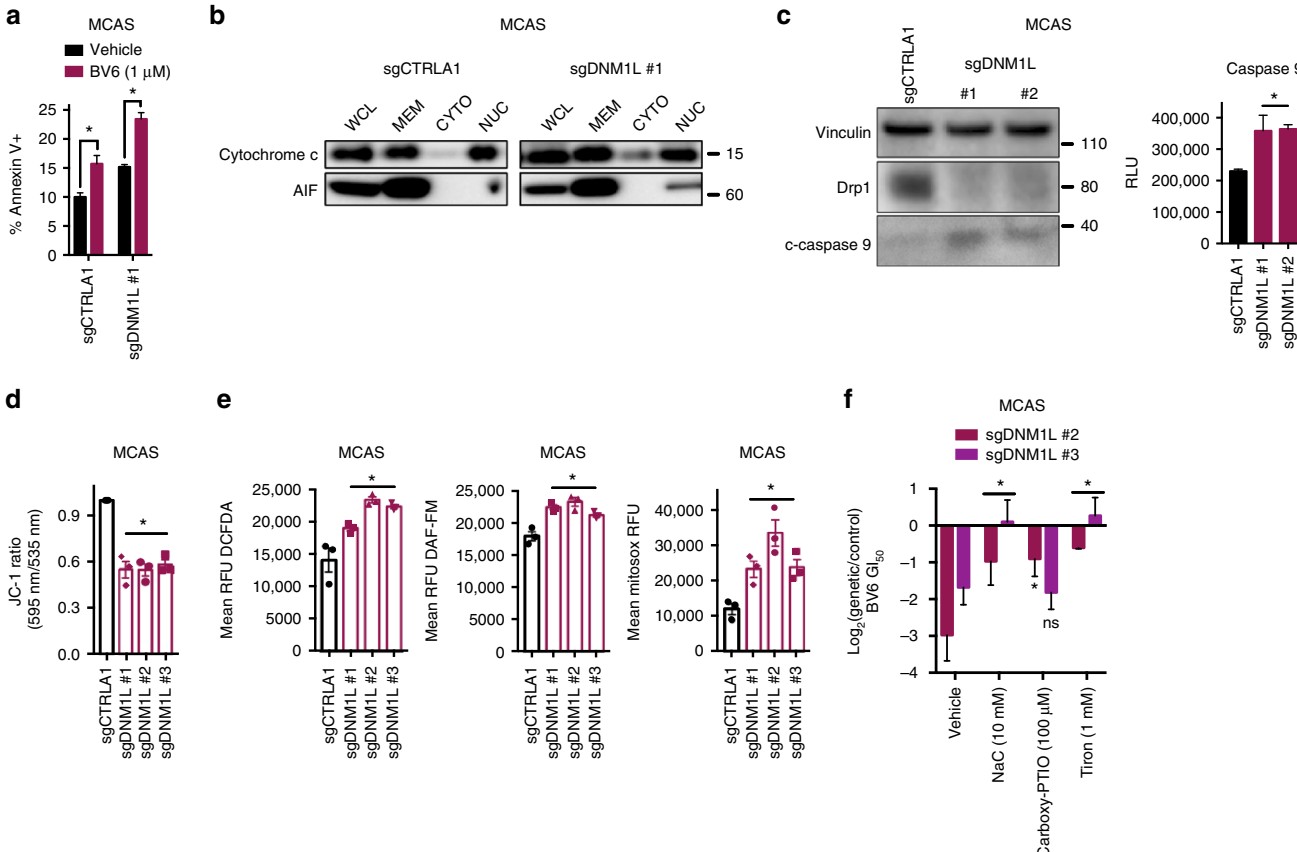

**Fig. 4** DNM1L loss leads to release of cytochrome c into the cytosol resulting in low-level caspase cleavage. **a** Percentage of annexin V+MCAS cells transduced with a control targeting sgRNA or a sgRNAs targeting DNM1L ($n = 3$). Data are mean ± SEM. *$p < 0.05$ by Student's $t$-test. **b** Immunoblot of cytochrome c or AIF in whole-cell lysate, membrane fraction, cytosolic fraction, and nuclear fraction of MCAS cells transduced with a control sgRNA or a representative sgRNA targeting DNM1L. Immunoblots are representative of three independent experiments. AIF is used as a control for contamination of the cytosolic fraction. **c** Left, immunoblot of vinculin, Drp1, and c-caspase 9 in MCAS cells transduced with control sgRNA or two independent sgRNAs for DNM1L. Immunoblots are representative of two independent experiments. Right, raw luminescence units of caspase 9 activity in MCAS cells transduced with control sgRNA or two independent sgRNAs for DNM1L ($n = 3$). Data are mean ± SEM. *$p < 0.05$ by one-way ANOVA. **d** Ratio of JC-1 fluorescence (595 nm/535 nm) in MCAS cells transduced with control sgRNA or three independent sgRNAs for DNM1L ($n = 3$). Data are mean ± SEM. *$p < 0.05$ by one-way ANOVA. **e** Left, DCFDA raw fluorescence units minus no stain fluorescence units in MCAS cells transduced with control sgRNA or three independent sgRNAs for DNM1L ($n = 3$). Data are mean ± SEM. *$p < 0.05$ by one-way ANOVA. Middle, DAF-FM raw fluorescence units minus no stain fluorescence units in MCAS cells transduced with control sgRNA or three independent sgRNAs for DNM1L ($n = 3$). Data are mean ± SEM. *$p < 0.05$ by one-way ANOVA. Right, mitosox raw fluorescence units minus no stain fluorescence units in MCAS cells transduced with control sgRNA or three independent sgRNAs for DNM1L ($n = 3$). Data are mean ± SEM. *$p < 0.05$ by one-way ANOVA. **f** Log$_2$(sgDNM1L/sgCTRLA1) BV6 GI$_{50}$ value in MCAS cells treated with either vehicle or one of the following: NaC (10 mM), Tiron (1 mM), Carboxy-PTIO (100 μM) ($n = 3$). Data are mean ± SEM. *$p < 0.05$ by one-way ANOVA

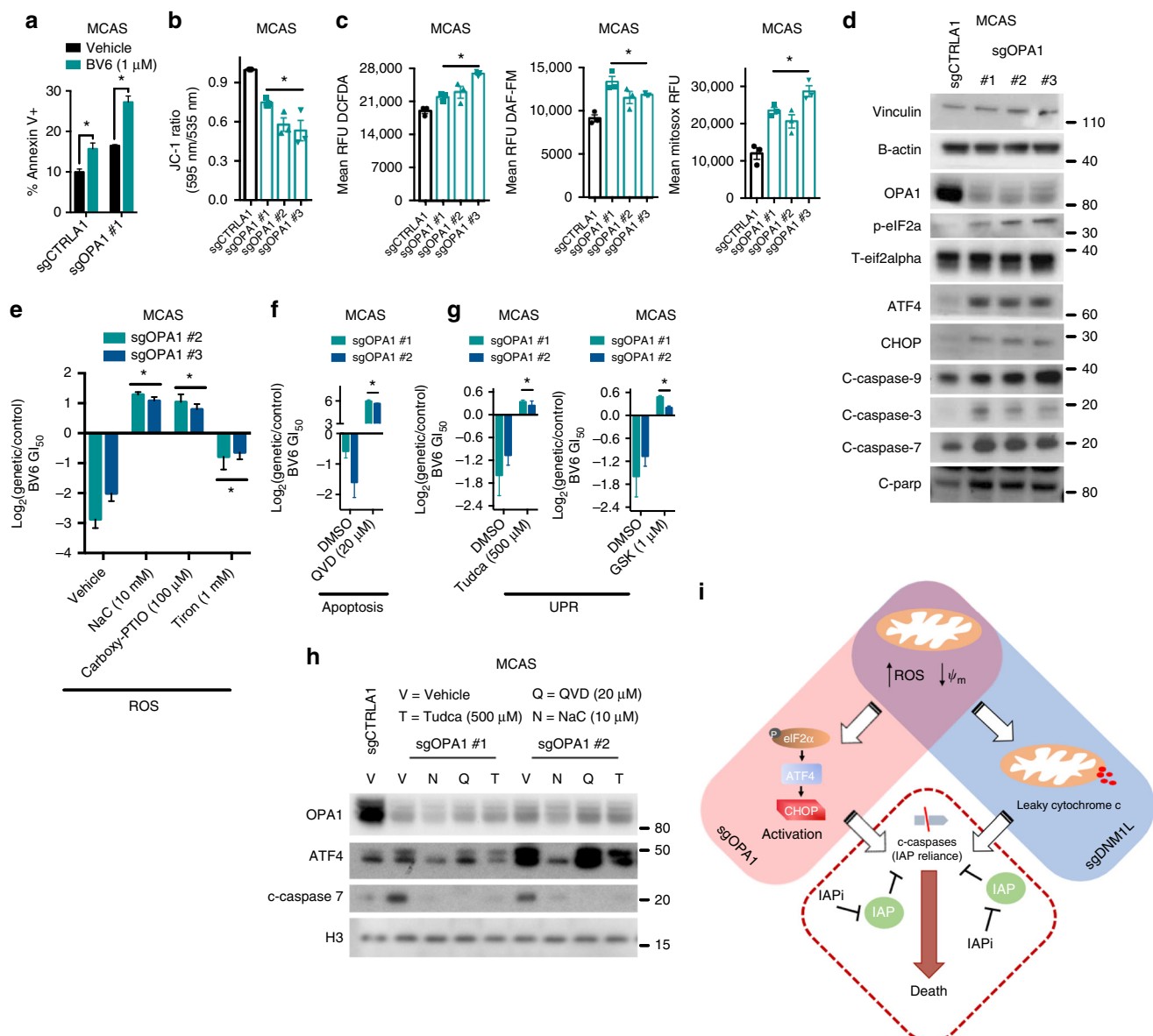

**Fig. 5** *OPA1* loss leads to induction of ER stress markers and changes in the levels of cleaved caspases. **a** Percentage of annexin V+MCAS cells transduced with a control targeting sgRNA or a sgRNAs targeting *OPA1* ($n = 3$). Data are mean ± SEM. *$p < 0.05$ by Student's t-test. **b** Ratio of JC-1 fluorescence (595 nm/535 nm) in MCAS cells transduced with control sgRNA or three independent sgRNAs for *OPA1* ($n = 3$). Data are mean ± SEM. *$p < 0.05$ by one-way ANOVA. **c** Left, DCFDA raw fluorescence units minus no stain fluorescence units in MCAS cells transduced with control sgRNA or three independent sgRNAs for *OPA1* ($n = 3$). Data are mean ± SEM. *$p < 0.05$ by one-way ANOVA. Middle, DAF-FM raw fluorescence units minus no stain fluorescence units in MCAS cells transduced with control sgRNA or three independent sgRNAs for *OPA1* ($n = 3$). Data are mean ± SEM. *$p < 0.05$ by one-way ANOVA. Right, mitosox raw fluorescence units minus no stain fluorescence units in MCAS cells transduced with control sgRNA or three independent sgRNAs for *OPA1* ($n = 3$). Data are mean ± SEM. *$p < 0.05$ by one-way ANOVA. **d** Immunoblot of indicated proteins from the UPR and apoptosis pathways in MCAS cells transduced with control sgRNA or three independent sgRNAs for *OPA1*. Immunoblots are representative of three independent experiments. **e** Log₂(sgOPA1/sgCTRLA1) BV6 GI₅₀ value in MCAS cells treated with either vehicle or one of the following: NaC (10 mM), Tiron (1 mM), and Carboxy-PTIO (100 μM) ($n = 3$). Data are mean ± SEM. *$p < 0.05$ by one-way ANOVA. **f** Log₂(sgOPA1/sgCTRLA1) BV6 GI₅₀ value in MCAS cells treated with either vehicle or Q-VD-OPh (20 mM) ($n = 3$). Data are mean ± SEM. *$p < 0.05$ by one-way ANOVA. **g** Log₂(sgOPA1/sgCTRLA1) BV6 GI₅₀ value in MCAS cells treated with either vehicle or one of the following: Tudca (500 μM) or GSK2606414 (1 μM) ($n = 3$). Data are mean ±SEM. *$p < 0.05$ by one-way ANOVA. **h** Immunoblot of indicated proteins in MCAS cells transduced with control sgRNA or two independent sgRNAs for *OPA1* treated with vehicle, QVD, Tudca, or NaC at indicated doses. Immunoblots are representative of two independent experiments. **i** Model of SMAC mimetic sensitivity in *OPA1* or *DNM1L* loss cells

the importance of apoptosis induction in therapeutic response[40–43]. Surprisingly, this sensitization to SMAC mimetics occurs through distinct molecular mechanisms in cells with increased fragmentation vs. increased connectivity: In the former, increased levels of UPR induction lead to sensitization via increased steady state activation of effector caspases,

while the latter, passive cytochrome c release from the mitochondria potentiates the toxicity of SMAC mimetics by increasing steady-state activation of initiator caspases. Finally, because oncogenes exert dominant control over mitochondrial dynamics homeostasis, oncogene targeted therapies sensitize tumors to SMAC mimetics via their effects on dynamics.

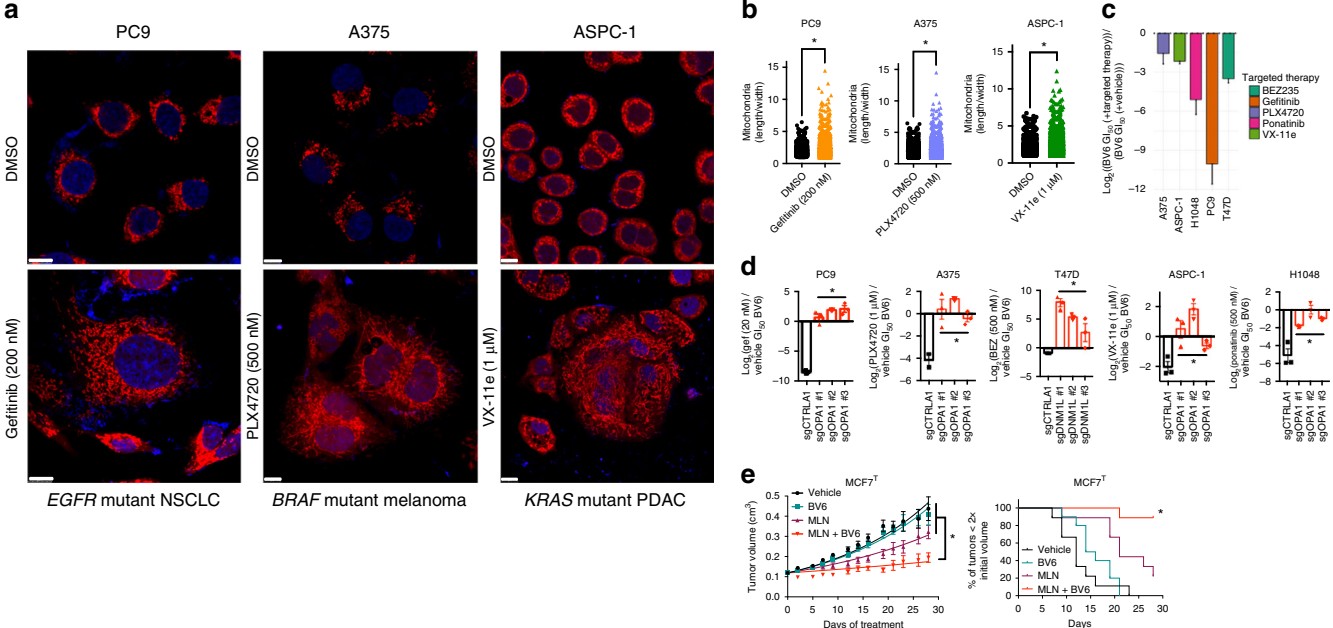

**Fig. 6** Oncogenic control over mitochondrial dynamics can be exploited therapeutically using targeted therapies to create a vulnerability to SMAC mimetics. **a** Mitotracker and DAPI in three cell lines PC9 (scale bars 10 μm), A375 (scale bars 7.5 μm), ASPC-1 (scale bars 10 μm) treated with vehicle or one of three inhibitors (gefitinib, PLX4720, VX-11e). Immunofluorescence images are representative of two independent experiments. **b** Quantification of images from Fig. 5a and Supplementary Fig. 5a. Mitochondrial length × width is plotted for thousands of mitochondria from >10 cells across at least two independent experiments. *$p < 0.05$ by Student's $t$-test. **c** $\log_2((\text{BV6 GI}_{50}\ (+\text{targeted therapy}))/(\text{BV6 GI}_{50}\ (+\text{vehicle})))$ in five cell lines driven by diverse oncogenes ($n = 3$). Data are mean ± SEM. **d** $\log_2((\text{BV6 GI}_{50}\ (+\text{targeted therapy}))/(\text{BV6 GI}_{50}\ (+\text{vehicle})))$ in five cell lines driven by diverse oncogenes transduced with a control sgRNA, three sgRNAs targeting DNM1L, or three sgRNAs targeting OPA1 ($n = 3$). Data are mean ± SEM. *$p < 0.05$ by one-way ANOVA. **e** Left, tumor volumes over time in an orthotopic model of advanced endocrine resistant breast cancer treated with vehicle, BV6, MLN-0128, or the combination (see Methods for $n$ in each group and statistics). Right, survival plot of in vivo data from Fig. 5d as the number of mice with tumors less than 2× the original size (see Methods for $n$ in each group and statistics). Vehicle and BV6 treated arms of the xenograft model presented here are the same data presented in Fig. 3h

Collectively, this work establishes altered mitochondrial dynamics as a targetable feature of human tumors.

This work motivates several avenues of future investigation. First, mechanistic studies to more precisely define the mechanisms by which alterations in dynamics-regulating genes lead to increased ROS and outer membrane depolarization are warranted, as are studies to understand how these properties lead to UPR induction in cells with increased fragmentation but cytochrome c release in cells with increased connectivity. Indeed, the literature points to additional mechanisms related to how structural changes in the mitochondria can affect apoptosis regulation in cells—mechanisms that could be working alongside those presented above[44]. Second, from a translational perspective, these studies motivate the testing of SMAC mimetics as anti-cancer therapies in patients, including both those patients with diseases like high-grade serous ovarian cancer that lack targetable driver oncogenes but frequently harbor amplifications in dynamics-regulating genes, and also in combination with approved targeted therapies. They also motivate efforts to create direct small-molecule inhibitors of mitochondrial fission/fusion proteins with improved pharmacological properties relative to the probe compound mDivi-1 for administration in human patients. Finally, these results suggest that a broad effort to further characterize the unique dependencies of tumors with dysregulated mitochondrial dynamics is likely to yield additional, selective therapeutic strategies.

Finally, and most broadly, this study demonstrates that selective tumor therapeutic targeting can be achieved by exploiting a dysregulated, macroscopic organelle property instead of a conventional oncogenic signaling or metabolic pathway. The structure, activity, and trafficking of many cellular organelles are altered in human tumors, and as such, this work may represent a starting point for many analogous studies.

## Methods

**Xenograft tumor study.** Eighty-four animals were estimated to be required to account for the take rate (anticipated 70% based upon prior experience), in order to provide at least 54 animals with tumors to be randomized for $n = 9$ or more per treatment. Six-week-old, female, athymic nu/nu mice were randomized by enrollment with animals being placed on treatment when tumor volume reached 0.10–0.15 cm³ volume. On each enrollment date, animals were allocated to treatment such that initial average tumor volumes for each group were equivalent. All procedures were approved by the Duke University Institutional Animal Care and Use Committee (IACUC) prior to initiating the experiment. Eighty-four female nu/nu mice (~6 weeks of age) were ovariectomized under anesthesia (isoflurane) and in the same procedure implanted sc (scapular region) with tamoxifen (Tam) treatment pellets (5 mg/60 days, ~3.3 mg/kg/d continuous release, Innovative Research of America) 24 h prior to having an ~8 mm³ section of tamoxifen-resistant (MCF7ᵀ) tumor tissue engrafted orthotopically (right axial mammary fat pad) under anesthesia. Tumors were measured 3× weekly, concurrent with weight and behavior monitoring, until tumors reached ~0.1–0.15 cm³ volume ($l × w^2 ×$ 0.5). Mice were then randomized ($n = 19$) to treatment with Vehicle (both oral and ip injection daily), mDivi-1 (15 mg/kg ip daily), or MLN-0128 (0.15 mg/kg oral gavage daily). mDivi-1 was formulated in 5% DMSO/95% PEG 400. MLN-0128 was formulated in 16% PVP/2.5% NMP. These three groups were further subdivided to receive ip injection of 2.5% DMSO/97.5% saline ($n = 9$) or BV6 (10 mg/kg dissolved DMSO and diluted in saline). Treatments were administered for 4 weeks with continued tumor measurement and behavioral monitoring. Personnel collecting primary data (tumor measurements and anima weights) and administering treatments were blinded to the study hypothesis. Safety precautions required for the treatments administered prevented blinding of personnel to treatment identities. Animals were then euthanized and blood and tissues retained for analyses. No animals were excluded from the analysis. Statistical analyses of animal studies were as follows: tumor growth data were subjected to exponential growth curve analysis constrained to share an initial value, and to two-way ANOVA

analysis followed by Bonferroni multiple comparison test. Significant difference as compared to the vehicle treated control ($p < 0.05$) was detected for multiple groups at several time points (indicated on graphs). Groups showed equivalent variance (10–15% with normal distribution) throughout all time points, justifying the statistical analyses that were selected.

**Cell lines and reagents**. All cell lines were grown at 37 °C in 5% $CO_2$. TYKNU, SKBR3 NCIH596, A375, PC9, MDAMB436, A549, HCC827, H1703, H2228, and H1048 were grown in RPMI 1640 supplemented with 10% FBS and 1% pen/strep. HCC1395, BT20, and MDAMB453 were grown in MEM supplemented with 5% NEAA, 5% pyruvate, 10% FBS, and 1% pen/strep. MCAS and CAL51 were grown in MEM supplemented with 20% FBS and 1% pen/strep. MCF7$^T$ was grown in DMEM/F12 supplemented with 5% glutamate, 5% NEAA, 10% FBS, and 1% pen/strep. TOV-112D, BT474, and T47D were grown in DMEM supplemented with 10% FBS and 1% pen/strep. Panc03.27, CALU-1, CALU-6, and APSC-1 were grown in DMEM/F12 supplemented with 10% FBS and 1% pen/strep. MCF10A was grown in MEBM (MEGM Kit without GA-1000) supplemented with cholera toxin. All cell lines were purchased from Duke University Cell Culture Facility (CCF) or given to us by Donald McDonnell (MCF7$^T$). All cell lines were authenticated using Promega PowerPlex 18D kit or were purchased within 6 months from Duke CCF. All cell lines were tested for mycoplasma by Duke CCF. Drugs were purchased from Selleck Chemicals, ChemieTek, MedChemExpress, Sigma-Aldrich, or APExBIO.

**Cloning CRISPR constructs**. CRISPR constructs were cloned following previous methods[46] using previously characterized sgRNAs[47]. sgRNA inserts were synthesized by IDT of the form:
    GGAAAGGACGAAACACCGXXXXXXXXXXXXXXXXXXXXGTTTTA-GAGCTAGAAATAGCAAGTTAAAATAAGGC
    "X" denotes unique 20mer sgRNA sequence (see "20-mer sequences" below).
    The oligo pool was diluted 1:100 in water and amplified using NEB Phusion Hotstart Flex enzyme master mix and the following primers:
    ArrayF: TAACTTGAAAGTATTTCGATTTCTTGGCTTTATA-TATCTTGTGGA
    AAGGACGAAACACCG
    ArrayR: ACTTTTTCAAGTTGATAACGGACTAGCCTTATTTTAACTTGC-TATTTCTAGCTCTAAAAC
    PCR protocol: 98 °C/30 s, 18 × [98 °C/10 s, 63 °C/10 s, 72 °C/15 s], 72 °C/3 min.
    Inserts were cleaned with Axygen PCR clean-up beads (1.8×; Fisher Scientific) and resuspended in molecular biology grade water. lentiCRISPRv2 (hygro) was digested with BsmBI (Thermo Fisher) for 2 h at 37 °C. The large ~13 kB band was gel extracted after size-selection on a 1% agarose gel. Using 100 ng of cut lentiCRISPRv2 and 40 ng of sgRNA oligos, a 20 μL Gibson assembly reaction was performed (30 min, 50 °C). After Gibson assembly, 1 μL of the reaction was transformed into electrocompetent Lucigen cells and spread on LB-ampicillin plates and incubated overnight. Single colonies were picked and underwent plasmid extraction using a Plasmid miniprep kit (Qiagen).
    20-mer sequences: sgCTRLA1: GTAGCGAACGTGTCCGGCGT
    sgOPA1 #1: CAAGTGGAATGACTTTGCGG
    sgOPA1 #2: ATACGCAAGATCATCTGCCA
    sgOPA1 #3: AGGAACTTTTAACACCACAG
    sgDNM1L #1: GAACCAGTTCCACACAGCGG
    sgDNM1L #2: GGGAGGGACCTGCTTCCCAG
    sgDNM1L #3: GGATTTGCCAGGAATGACCA

**Lentivirus production and transduction**. HEK 293 T cells were grown in 15 cm to ~50% confluence. For each plate, transfection was performed using Fugene6 (Promega), 6.2 μg of psPAX2, 0.620 μg pVSVg, 6.25 μg of CRISPR plasmid. After 30 min of incubation at room temperature, the mixture was added to the cells and incubated overnight. The next day harvest media was added (DMEM 30% FBS). After two collections at 24 h each, the harvested virus was passed through a 0.45 μm filter. Transductions were performed by seeding cells at ~40% confluence into six-well dishes, then the following day adding 0.5 mL of virus, 0.5 mL of media, and 2 μL polybrene to the cells. The cells were then centrifuged to the following specifications: 1 h, 4 °C, 2500 RPM. Following the spin, fresh media without virus or polybrene was placed onto the cells. The following day, cells were selected with appropriate selection antibiotic.

**shRNA DNA prep and constructs**. shRNA glycerol stocks were obtained from the Duke Functional Genomics Core Facility. Glycerol stocks were streaked out on LB/Amp plates overnight. The following day colonies were picked and grown overnight in liquid culture. The next day, the DNA was prepped using a Qiagen Miniprep kit. DNA was subsequently used to make lentivirus. Scramble shRNA was a gift from David Sabatini (Addgene plasmid # 1864). The identity, TRC number, and sequences of the hairpins are listed below:
    shXIAP #1 TRCN0000003785
GAGCTGTAGATAGATGGCAATACTCGAGTATTGCCATCTATCTACA
    shXIAP #2 TRCN0000003786
GGCACTCCAACTTCTAATCAAACTCGAGTTTGATTAGAAGTTGGAGT

    shCIAP1 #1 TRCN0000003780
GGCCGAATTGTCTTTGGTGCTTCTCGAGAAGCACCAAAGACAATTCG
    shCIAP1 #2 TRCN0000003782
GGCTGCGGCCAACATCTTCAAACTCGAGTTTGAAGATGTTGGCCGCA
    shCIAP2 #1 TRCN0000003776
GGCACTACAAACACAATATTCACTCGAGTGAATATTGTGTTTGTAGT
    shCIAP2 #2 TRCN0000003779
GGCTCTTATTCAAACTCTCCATCTCGAGATGGAGAGTTTGAATAAGA
    shScramble N/A
CCTAAGGTTAAGTCGCCCTCGCTCGAGCGAGGGCGACTTAACCTTAGG

**Short-term growth-inhibition assay**. Cells were seeded into 96-well plates at 2000 cells/well. To generate GI$_{50}$ curves, cells were treated with vehicle (DMSO) or an eight-log serial dilution of drug. Each treatment condition was represented by at least three replicates. Three days after drug addition, cell viability was measured using Cell Titer Glo® (Promega). Relative viability was then calculated by normalizing luminescence values for each treatment condition to control treated wells. To generate GI$_{50}$ curves for drug combinations, slight modifications are made. Primary drug was applied and diluted as above while the second drug was kept at a constant concentration across all wells except the DMSO-only condition. Viability for all primary drug dilutions was then calculated relative to luminescence values from the secondary drug-only condition. We plot the viability vs. concentration curve for drug A (normalized appropriately to the viability of cells treated with DMSO in media control). Next, we plot the viability vs. concentration curve for drug A in the presence of a fixed dose of drug B (this time normalizing to the viability of cells treated with drug B alone). Sensitization of cells to drug A by drug B is evidenced by a leftward shift in the curve. Dose–response curves were fit using Graph pad/ Prism 6 software.

**Western blotting and antibodies**. Immunoblotting was performed as previously described[45] and membranes were probed with primary antibodies (1:1000 dilution) recognizing vinculin (CST#4650), H3 (CST#4499), Bim (CST#2933), Bid (CST#2002), BCL-X$_L$ (CST#2764), β-Actin (CST#4970), Opa1 (CST #80471), Drp1 (CST #5391), cytochrome c (CST #11940), p-IRE1 and T-IRE1 (Abcam #124945, Abcam #37073), Bax (CST#5023), XBP-1s (CST #12782), ATF6 (CST #65880), ATF3 (Abcam #207434), BCL2 (CST #2870), BCL-w (CST #2724), ATF4 (CST #11815), AIF (CST #5318), CHOP (CST #5554), c-caspase 3 (CST #9664), c-caspase 7 (CST #9491), c-caspase 9 (CST #9501), c-caspase 8 and caspase 8 (CST #9748 and CST #4790), p-eIF2α and T- eIF2α (CST #3398 and CST #5324), c-parp (CST #9546), Puma (CST #12450), and X-IAP (CST #2045). Briefly, cells were resuspended in lysis buffer, incubated on ice for 15 min, then clarified at 13,000 RPM, 4 °C, for 10 min. Protein was quantified using the Bradford method and lysates were made with NuPage Sample Buffer (4×). For cell fractionation studies, the Cell Fractionation Kit (CST #9038) was used following the manufacturer's instructions. For all representative immunoblots in the manuscript, experiments were conducted at least twice, and had no repeatability issues. For IF studies, the following primary and secondary antibodies were used: Cytochrome c (CST #12963), Tom20, (CST #42406), Anti-rabbit Alexa 488 (CST #4412), Anti-mouse Alexa 594 (CST #8890). For all representative images in the manuscript, experiments were conducted at least twice, and had no repeatability issues. Uncropped western images for main text figures can be found in Supplementary Fig. 8.

**Chemical library screening**. The ~2100 compound library was screened in duplicate at 2 and 10 μM. TYKNU control cells or each of the two genetic derivatives (OPA1 or DNM1L) were seeded into drug-stamped 384-well plates at 500 cells per well. After 72 h of drug-treatment the assay was read using Cell Titer Glo. Duplicate treatment wells were averaged and normalized to duplicate control wells from the same plate position. Normalized values were then processed by calculating the log$_2$ (genetic derivative/genetic control) for each compound at both doses. Then we impose a threshold of ~2.5 standard deviations away from the mean value for sensitive hits (log$_2$ < −0.5) and a threshold of ~3.5 standard deviations away from the mean value for resistant hits (log$_2$ > 0.7) using the distribution of the control well values.

**Quantification of apoptosis by annexin-V**. Cells were seeded in six-well plates and treated the next day with either the indicated amount of drug, vehicle (DMSO), or combination. Cells were incubated for 2 days, washed twice with ice-cold PBS, and resuspended in 1× annexin V binding buffer (10 mM HEPES, 140 mM NaCl, 2.5 mM $CaCl_2$; BD Biosciences). Surface exposure of phosphatidylserine was measured using APC-conjugated annexin V (BD Biosciences). 7-AAD (BD Biosciences) or PI (Thermo) was used as a viability probe. Experiments were analyzed at 20,000 counts/sample using BD FACSVantage SE. Gatings were defined using untreated/unstained cells as appropriate.

**Fluorescence microscopy**. Cell lines were plated on glass coverslips and the following day were treated with 100 nM MitoTracker Red CMXRos (Life Technologies) for 30 min, fixed (formaldehyde), permeabilized (Triton-X), and mounted using Prolong Gold anti-fade reagent with DAPI (Life Technologies). For IHC fluorescence microscopy, protocol was slightly altered the day after plating. Cells

were washed in PBS then put in blocking buffer (PBS, 0.4% Triton-X-100, 2% normal goat serum, 2% BSA) at room temperature for 30 min. Cells were washed in PBS, and primary antibody (diluted in blocking buffer) was added to the cells overnight at 4 degrees C. The next day, cells were washed three times with PBS. Secondary antibody (diluted in blocking buffer) was added to cells for 90 min at room temperature. Finally, the cells were washed three times with PBS and mounted as listed above. The slides were then imaged using a Leica SP5 inverted confocal microscope with ×40 oil objective. For all representative images in the manuscript, experiments were conducted at least twice, and had no repeatability issues. Mitochondrial morphology was determined using the tubeness and vesselness algorithms in Fiji. The analysis methodology was optimized on an unrelated and independent set of images and applied across all images obtained for this study. Mitochondrial fragmentation vs. connectivity was determined by plotting length × width of several thousand mitochondrial from at least 10 cells across at least two independent experiments. For co-localization analysis, Fiji Coloc2 analysis software was used to determine the Pearson correlation of two different image channels in at least 25 cells across at least two independent experiments. Higher correlation values indicate a greater extent of co-localization.

**Caspase-9 activity**. Caspase-9 activity was determined using Caspase-Glo® 9 Assay (Promega) according to the manufacturer's recommendations.

**Caspase-8 activity**. Caspase-8 activity was determined using Caspase-Glo® 8 Assay (Promega) according to the manufacturer's recommendations.

**Membrane potential assay**. Membrane potential was determined in 96-well format using JC-1 Mitochondrial Membrane Potential Assay Kit (Cayman Chemical) according to manufacturer recommendations.

**ROS detection**. ROS was detected using MitoSOX™ Red (Thermo Fisher), DAF-FM diacetate, or DCF-DA. Cells were collected and incubated with 5 μM MitoSOX, DAF-FM, or DCF-DA for 30 min then washed with PBS. After washing, cells were incubated in PBS for an additional 20 min, then analyzed using microplate fluorescent detection on a Tecan Infinite M1000 reader.

**Statistical analysis**. Unless otherwise specified, Student's $t$-tests, or for grouped analyses, one-way ANOVA with Tukey's post hoc test were performed and $p$ values < 0.05 were considered significant. Results are presented as means ± SEM.

**Data availability**. All relevant data are available within the manuscript and its supplementary information or from the authors upon reasonable request. Please contact Kris Wood (Kris.Wood@duke.edu) or Gray Anderson (grayranderson@gmail.com) for any requests. The Cancer Genome Atlas data referenced during the study are available in a public repository from the https://cancergenome.nih.gov website. The Genomics of Drug Sensitivity data referenced during the study are available in a public repository from the http://www.cancerrxgene.org website. The Cell Line Encyclopedia data referenced during the study are available in a public repository from the https://portals.broadinstitute.org/ccle website.

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

## Acknowledgements

We thank the members of the Wood and McDonnell laboratories for helpful discussions and technical assistance. We also thank the faculty and students in the Department of Pharmacology and Cancer Biology at Duke University for helpful feedback during the annual scientific retreat. Finally, we thank Stacy Horner (Assistant Professor, Duke University) and Christopher Nicchitta (Professor, Duke University) for helpful discussions. This work was supported by Duke University School of Medicine start-up funds, the NIH Building Interdisciplinary Research Careers in Women's Health (BIRCWH) Program (K12HD043446), a Liz Tilberis Early Career Award from the Ovarian Cancer Research Fund Alliance, a Department of Defense Breast Cancer Research Program Breakthrough Award (W81XWH-16-1-0703), and NIH award R01CA207083 (all to K.C. W.). This work was also supported by the National Science Foundation Graduate Research Fellowship Program (DGE-1106401 to G.R.A.) and a NIH/NCI Pre-to-Postdoctoral Transition Award (1F99CA222728 to G.R.A.). Any opinions, findings, and conclusions or recommendations expressed in this material are those of the authors(s) and do not necessarily reflect the views of the NIH or NSF.

## Author contributions

Conceptualization: G.R.A. and K.C.W.; methodology: K.C.W., G.R.A., S.E.W., M.C., and D.P.M.; formal analysis: K.C.W., G.R.A., M.C., and S.E.W.; investigation: G.R.A., S.E.W., M.C., C.Y., Y.A., M.A., A.P.Y., and C.A.C.; writing of original draft: K.C.W. and G.R.A.; writing, review, and editing: all authors; funding acquisition: K.C.W. and D.P.M.; resources: K.C.W. and D.P.M; supervision: K.C.W. and D.P.M.

## Additional information

**Competing interests:** The authors declare no competing interests.

