## [Peer Review File · Nature Communications]

Reviewers' comments:

Reviewer #1 (Remarks to the Author):

Manuscript NCOMMS-17-30156-T

Dysregulated mitochondrial dynamics are a targetable feature of human tumors

By Anderson et al.

The authors investigated the relevance of dysregulated mitochondrial dynamics by a disturbed balance in mitochondrial fission or mitochondrial fusion as a regulation of sensitivity of human cancers to therapeutic targeting. They report that genes regulating mitochondrial dynamics are frequently amplified in human cancers, imposing a vulnerability that can be therapeutically exploited. The authors found that tumor with increased mitochondrial fission or fusion are particularly sensitive to Smac mimetics and analyzed in further detail the underlying molecular mechanisms.

General comments:

While the authors report an interesting correlation between dysregulated mitochondrial dynamics and sensitivity of cancer cells to Smac mimetics, the underlying mechanisms responsible for this specific sensitivity have not yet been conclusively elucidated. It has also not been established that disturbed mitochondrial dynamics selectively sensitize for Smac mimetics and not for other apoptosis-inducing drugs. Furthermore, the links between the suggested pathways, in particular ER stress response and induction of cell death, remain obscure.

Major points:

Figure 3:

To explore whether cancer cells with increased mitochondrial fission or fusion are particularly susceptible to inhibition of IAP proteins the authors should employ a genetic approach in addition to the pharmacological approach by using Smac mimetics. To this end, the authors should examine the effect of genetic silencing of XIAP, cIAP1 and cIAP2 alone and in combination in cancer cells with disturbed mitochondrial fission or fusion. Furthermore, screening results displayed in panel A should be confirmed by using a second assay which directly determines induction of cell death. Also, the question arises whether silencing of OPA1 or DNMI1L selectively sensitizes cancer cells to Smac mimetics or whether it confirms a broader increased sensitivity to apoptosis-inducing drugs. To address this question additional apoptosis-inducing drugs should be tested in OPA1 and DNMI1L knockout cells. As far as experiments using mDivi-1 are concerned, these experiments should be confirmed by a genetic strategy to downregulate DRP1, since mDivi-1 has been reported to exert also DRP1-independent effects. This applies not only to cellular in vitro studies but also to the in vivo studies displayed in panel H of figure 3. Here, the effect of DRP1 knockdown or knockout on tumor growth alone and together with Smac mimetics should be tested. For in vivo experiments, it will also be important to extend the observation period beyond four weeks, since data displayed in panel H of figure 3 show that the tumor-suppressive effects are only minor after this short observation period.

Figure 4:

Since there is only a minor increase in apoptotic cells upon treatment with MDivi-1 and Smac mimetics, although this effect is statistically significant, the question arises as to whether cells indeed die via apoptosis or whether other forms of programmed cell death might also be involved. To test this in a first approach it is suggested to perform Annexin V/propidium iodide (PI) double

staining and to present data for all four different quadrants, i.e. Annexin V-positive/PI-negative cells, Annexin V- and PI-double positive cells, Annexin V-negative/PI-positive cells and cells negative for both Annexin V and PI. In panel B of figure 4, controls are lacking showing the purity of this cytosolic fraction without mitochondrial contaminations. As far as panel C of figure 4 is concerned, the data show only a minor increase in cleaved caspase-9 and a minor increase in caspase-9 enzymatic activity, casting doubt about the relevance of caspase-9 activation in this context. Additional assays should therefore be performed to address the question whether caspase-9 becomes activated or not. Furthermore, the specificity of caspase-9 activation is unknown at present. Therefore, the experiments should be extended to caspase-8 activation with an additional initiator caspase besides caspase-9. As far as panel D of figure 4 is concerned, the authors should employ additional fluorogenic dyes besides Mitosox to determine generation of ROS. Similarly, additional ROS scavengers besides NAC should be used to confirm the involvement of ROS in this context, since NAC lacks specificity. To determine which branch of the ER stress response is activated, the experiments need to be extended to additional parameters of ER stress response, e.g. phosphorylation of PERK and XBP-1 splicing. In panels H to G, additional ROS scavengers besides NAC should be used. As far as Tudca is concerned, controls are lacking to demonstrate that this compound interferes with the unfolded protein response and ER stress induction.

Reviewer #2 (Remarks to the Author):

This study from Anderson et al. seeks to understand how alterations in mitochondrial dynamics affect therapeutic vulnerabilities in a wide variety of human tumors. Using a large scale screening strategy in tumor cell lines the authors identify that shifting the mitochondrial network in either direction impacts sensitivity to SMAC mimetics and propose two different models to explain these results. This paper tackles a very important and emerging area in cancer biology and reveals some very interesting findings. However, there are a number of issues with both study design and data interpretation that lessen my enthusiasm for this work. Most notably, an over-reliance on Mdivi-1, a drug with well known off target effects on ETC function, makes it difficult to interpret much of the pharmacological data. Also, there is a failure to consider alternative and quite plausible mechanisms through which Opa1 inhibition might be affecting apoptosis sensitivity. Specific concerns are listed below:

In the abstract the authors claim: "...there have been no studies to define therapeutic vulnerabilities resulting from these alterations." and in the intro they claim: "Despite the observation that mitochondrial dynamics are frequently altered in human cancers and the likelihood that these alterations broadly impact cell physiology, there have been no efforts to define therapeutic vulnerabilities driven by altered dynamics." - This is an interesting study and their approach has potential to provide valuable insights, but this claim seems to be an unnecessary attempt to claim novelty. There are dozens of studies, including several of those cited in the manuscript (E.g. - Ref. #4. Also... Qian, et al. Novel combination of mitochondrial division inhibitor 1 (mdivi-1) and platinum agents produces synergistic pro-apoptotic effect in drug resistant tumor cells. *Oncotarget* (2014), and Han, X.-J. et al. Mitochondrial dynamics regulates hypoxia-induced migration and antineoplastic activity of cisplatin in breast cancer cells. *Int. J. Oncol.* (2014)), that directly address the question of how changes in mitochondrial dynamics affect therapeutic vulnerability in cancer and several of these studies have made important insights into potential mechanisms through which these vulnerabilities arise.

Figure 1 - The literature provides plenty of rationale to investigate how alterations in mitochondrial fusion/fission dynamics affect sensitivity to drug treatments. In figure 1, the authors mine publicly available data to provide additional rationale and to begin to identify potential vulnerabilities. There are, however, several issues with the data presented here. Most notably, several of the genes analyzed in this figure reside in genomic regions with other genes whose amplification can

explain the differences in drug sensitivity as well or better than the gene being highlighted. For example, *Mfn1*, the most highly amplified gene in figure 1A, lies in a region of chromosome 3 within 150 kilobases of *PIK3CA*. Similarly, *DNM1L* is near the *KRAS* gene on chromosome 12, the amplification of which is validated to play a role in cancer and may better explain the differential sensitivity to PI3K pathway inhibitors. To draw conclusions about the potential relationship between mitochondrial dynamics machinery amplification and drug sensitivity from this data, the authors would need to control for amplification of nearby oncogenes in each of these regions to see if the correlations that have been identified are robust.

Figure 3 - It has long been appreciated that Mdivi-1 has numerous off target effects and a recent study demonstrates both that it is a Complex I inhibitor and potentially a poor Drp1 inhibitor (Bordt et al Dev Cell. 2017). For this reason, the genetic approaches presented in this figure are much more compelling than the pharmacological approaches. Complex I inhibition by Mdivi-1 will have a lot of downstream effects (see the literature on metformin) so unfortunately, this drug is not a very useful tool for understanding the biology of mitochondrial fission.

Figure 4 - The data arguing for increased sensitivity to SMAC mimetics due to leakiness of cytochrome C are not very robust and this model is not well supported. The western blots in 1B need some controls (mitochondrial proteins to show purity of cytoplasmic fraction, e.g.). Also, the western data as presented (both cyt. C and caspase 9) is consistent with incomplete MOMP (iMOMP) in a large fraction of the cells or complete MOMP in a small fraction of the cells. I.F. would be able to distinguish these possibilities, the former of which would be consistent with the cyt. C leakiness proposed by the authors.

Figure 4 - The data on Opa1 are compelling, but the model ignores an additional aspect of Opa1 biology that might explain the data. Opa1 is important for maintenance of cristae junctions and its cleavage is important for complete cytochrome c release, as much of the cytochrome c can otherwise be trapped in the cristae folds. It is possible that Opa1 deletion is allowing for cytochrome c release (but maybe not IAP release) from mitochondria under conditions where it is normally prevented (ie - Bax/Bak independent damage to OMM). This would also explain the sensitivity to SMAC mimetics and is consistent with a known role for Opa1.

Additional comments:

The authors consistently write that Opa1 inhibition increases mitochondrial fission and Drp1 inhibition increases fusion. They should be more careful with the language here and use more precise wording such as "fragmentation" and "connectivity". Fusion activity is not directly affected by inhibition of fission nor is fission activity affected by fusion inhibition.

For images of mitochondrial morphology, the authors show limited replicates and perform no quantitation of the observed phenotypes.

It is never examined whether there is any relationship between the copy number alterations and mitochondrial morphology, even for a small subset of available cell lines. This would strengthen the arguments made in the paper, especially for figure 1 and 3D.

Response to Reviewers' comments:

Reviewer #1: Thank you for your helpful remarks and suggestions, which we believe have substantially improved the manuscript. Please see below for answers to each question raised in your initial review of our manuscript.

Q1: While the authors report an interesting correlation between dysregulated mitochondrial dynamics and sensitivity of cancer cells to Smac mimetics, the underlying mechanisms responsible for this specific sensitivity have not yet been conclusively elucidated.

A1: We thank the reviewer for the insightful comments, below, related to the mechanism, and we ask that he or she consider our responses to each specific comment.

Q2: It has also not been established that disturbed mitochondrial dynamics selectively sensitize for Smac mimetics and not for other apoptosis-inducing drugs.

A2: The reviewer is astute in noting that disturbed mitochondrial dynamics could potentially sensitize cells to a broader class of apoptosis-inducing drugs. Indeed, in our screening data, the dual BCL-2/BCL-X_L inhibitor (ABT737, ABT263) scored as a robust hit, suggesting that other targeted apoptosis-inducing drugs may have increased potency in the context of perturbed dynamic states (Figure 3A). We have included validation of ABT737 as a hit and have also validated the increased sensitivity of cells with disturbed mitochondrial dynamics to another targeted apoptosis-inducing BH3 mimetic, the MCL-1 inhibitor S63845. Further, we were able to show that a general apoptosis-inducing drug, etoposide, did not exhibit increased potency in cells with perturbed dynamics (Supplemental Figure 3I).

Q3: Furthermore, the links between the suggested pathways, in particular ER stress response and induction of cell death, remain obscure.

A3: We thank the reviewer for his or her comments related to the ER stress phenotype and apoptosis and ask that the reviewer refer to our detailed responses to specific mechanistic questions below.

Q4: To explore whether cancer cells with increased mitochondrial fission or fusion are particularly susceptible to inhibition of IAP proteins the authors should employ a genetic approach in addition to the pharmacological approach by using Smac mimetics. To this end, the authors should examine the effect of genetic silencing of XIAP, cIAP1 and cIAP2 alone and in combination in cancer cells with disturbed mitochondrial fission or fusion.

A4: We thank the reviewer for pointing out that we did not include genetic validation of the SMAC mimetic targets. We have now used genetic knockdown of *XIAP*, *CIAP1*, or *CIAP2* alone and in combination with genetic knockout of *OPA1* or *DNM1L*, demonstrating that cells with perturbed mitochondrial dynamics exhibit increased sensitivity to knockdown of each of the three targets of SMAC mimetics listed above (Supplemental Figure 3A).

Q5: Furthermore, screening results displayed in panel A should be confirmed by using a second assay which directly determines induction of cell death.

A5: We thank the reviewer for suggesting a secondary validation of our phenotype that directly detects apoptotic cell death. We have now included annexin V and PI staining for cells with *OPA1* or *DNM1L* knockout treated with vehicle or a SMAC mimetic. We see that the *OPA1* or *DNM1L* knockout cells have higher levels of annexin V+ (and annexin V+ / PI-) staining than the control knockout cells. Further, we also find that SMAC mimetics induce a higher degree of annexin V positivity (and annexin V+ / PI- staining) in the *OPA1* or *DNM1L* knockout cells than the control cells. (Figure 4A, Figure 4G, and Supplemental Figure 4A).

Q6: Also, the question arises whether silencing of *OPA1* or *DNM1L* selectively sensitizes cancer cells to Smac mimetics or whether it confirms a broader increased sensitivity to apoptosis-inducing drugs. To address this question additional apoptosis-inducing drugs should be tested in *OPA1* and *DNM1L* knockout cells.

A6: We ask that the reviewer please see our answer to this question above in answer 2.

Q7: As far as experiments using mDivi-1 are concerned, these experiments should be confirmed by a genetic strategy to downregulate DRP1, since mDivi-1 has been reported to exert also DRP1-independent effects. This applies not only to cellular *in vitro* studies but also to the *in vivo* studies displayed in panel H of figure 3. Here, the effect of DRP1 knockdown or knockout on tumor growth alone and together with Smac mimetics should be tested.

A7: We agree with the reviewer that off-target effects are always a concern when using chemical probes like mDivi-1, a tool compound used to block Drp1. For this reason, for all experiments in the study with the exception of the *in vivo* data, we have confirmed that *DNM1L* knockout phenocopies the effects of mDivi-1 treatment (Figures 2-5). Further, we point out that all of these genetic experiments were actually included in the originally submitted manuscript's supplemental figures. In the revised manuscript, we have rearranged the presentation of these data to make them more apparent to the reader, and we have also modified the text to highlight the potential for off-target effects associated with mDivi-1.

To build on this genetic phenocopying data, we have also experimentally shown that mDivi-1 treatment fails to further sensitize cells with *DNM1L* knockout to SMAC mimetics, suggesting that the effects of mDivi-1 treatment are mediated by on-target Drp1 inhibition (Supplemental Figure 3H). Lastly, it is worth mentioning that our study uses doses of mDivi-1 in the range of 10-25 μ M, consistent with those used to perturb mitochondrial morphology in the original publication describing this compound (Cassidy-Stone et al. 2008). By contrast, evidence of ETC dysfunction resulting from mDivi-1 treatment was observed at higher doses (50-150 μ M) (Bordt et al. 2017).

Finally, it is unfortunately infeasible to repeat the *in vivo* studies using *DNM1L* knockout in a timely manner; such studies would require another 6-12 months because of inherent technical challenges. Specifically, *DNM1L* knockout is slightly toxic to cells, slowing their growth rate relative to wild-type cells. As a result, in populations of cells in which *DNM1L* has been knocked out, minority clones with incomplete knockout overtake the population within a

relatively short timeframe. Thus, performing a xenograft study in this configuration would likely fail on technical grounds because of the re-emergence of Drp1-expressing cells on the same timescale as tumor formation itself. A solution to this challenge is to use single cell cloning to obtain clonal populations of *DNM1L*^{-/-} cells, then perform xenograft studies with them. However, this procedure takes months to perform. Further, it would necessitate the use of many distinct *DNM1L*^{-/-} tumors, derived from distinct cellular clones, to ensure that the effects observed are associated with *DNM1L* loss and not random clonal variation. Together, the timeframe and economic constraints associated with doing genetic knockout studies *in vivo* preclude us from performing them, and we ask that the reviewer consider our extensive *in vitro* data as an appropriate alternative to this experiment, particularly given the likely cell autonomous nature of SMAC mimetic toxicity in tumors with altered mitochondrial dynamics.

Q8: For *in vivo* experiments, it will also be important to extend the observation period beyond four weeks, since data displayed in panel H of figure 3 show that the tumor-suppressive effects are only minor after this short observation period.

A8: We agree with the reviewer that the tumor-suppressive effects are statistically significant but relatively modest in the four-week period of the experiment presented in this figure. However, we point out that this experiment was simply used to demonstrate that pharmacological inhibition of Drp1 is sufficient to sensitize tumors to SMAC mimetic therapy *in vivo* with minimal toxicity, in the absence of extensive pharmacokinetic and pharmacodynamic optimizations. We feel the existing tumor size data, survival data, and mouse weights firmly support this concept. Further, in the revised text we underscore the proof-of-concept nature of this experiment, and further use it to highlight the fact that more potent and selective inhibitors of mDivi-1, optimized with respect to pharmacokinetic and pharmacodynamics properties, may produce even more substantial tumor growth inhibition in the context of SMAC mimetic therapy.

Q9: ...suggested to perform Annexin V/propidium iodide (PI) double staining and to present data for all four different quadrants, i.e. Annexin V-positive/PI-negative cells, Annexin V- and PI-double positive cells, Annexin V-negative/PI-positive cells and cells negative for both Annexin V and PI.

A9: This is an excellent suggestion, and these data will provide evidence of potential non-apoptotic cell death effects at play. We have performed the experiment requested and see that we are able to detect Annexin V+/PI- and Annexin V+/PI+ cells, but we are unable to detect Annexin V-, PI+ cells (Supplemental Figure 4A).

Q10: In panel B of figure 4, controls are lacking showing the purity of this cytosolic fraction without mitochondrial contaminations.

A10: The reviewer is correct to point out that controls are necessary in order to demonstrate that cell fractionation experiments were performed appropriately. We thank the reviewer for pointing this out and have added an immunoblot against AIF as a control in panel B of Figure 4.

Q11: As far as panel C of figure 4 is concerned, the data show only a minor increase in cleaved caspase-9 and a minor increase in caspase-9 enzymatic activity, casting doubt

about the relevance of caspase-9 activation in this context. Additional assays should therefore be performed to address the question whether caspase-9 becomes activated or not. Furthermore, the specificity of caspase-9 activation is unknown at present. Therefore, the experiments should be extended to caspase-8 activation with an additional initiator caspase besides caspase-9.

A11: We thank the reviewer for suggesting that we take a look at the activation of caspase 8. We show through immunoblotting that we are unable to detect c-caspase 8 (Supplemental Figure 4F). In addition, we confirmed this with a secondary luminescence-based assay of caspase 8 activity (Supplemental Figure 4F). Furthermore, we provide additional support for our model of leaky cytochrome c in Supplemental Figure 4G. We ask the reviewer to please see our response to Reviewer #2 Q7.

Q12: The authors should employ additional fluorogenic dyes besides Mitosox to determine generation of ROS.

A12: Since the original submission, we have added two additional fluorogenic dyes (DAF-FM Acetate and DCF-DA) besides Mitosox to detect ROS in *OPA1* and *DNM1L* knockout cells (Figure 4E and Figure 4I). The results are consistent with our findings with Mitosox.

Q13: Similarly, additional ROS scavengers besides NAC should be used to confirm the involvement of ROS in this context, since NAC lacks specificity.

A13: Since the original submission, we have added two additional ROS scavengers (tiron and carboxy-PTIO) besides NaC to rescue ROS in *OPA1* and *DNM1L* knockout cells (Figure 4F and Figure 4K). The results are consistent with our findings with NaC.

Q14: To determine which branch of the ER stress response is activated, the experiments need to be extended to additional parameters of ER stress response.

A14: We have immunoblotted *OPA1* knockout cells for the three major branches of the UPR, including for p-eIF2 α , ATF4, CHOP, XBP-1s, p-IRE1, ATF3, and ATF6 (Figure 4J, Supplemental Figure 4K). We see that the eIF2 α -ATF4-CHOP arm of the UPR is the only arm consistently activated in *OPA1* knockout cells (Figure 4J, Supplemental Figure 4K). Further, we show functional relevance of that specific arm through the addition of a PERK inhibitor (GSK2606414) which is also able to rescue the sensitivity to BV6 (Figure 4M). We note that all of these data were presented in the original submission.

Q15: In panels H to G, additional ROS scavengers besides NAC should be used.

A15: Please see the answer to Q13.

Q16: As far as Tudca is concerned, controls are lacking to demonstrate that this compound interferes with the unfolded protein response and ER stress induction.

A16: ATF4 is primary effector of ER stress observed in *OPA1* knockout cells (Figure 4J). We have demonstrated that treatment with Tudca can prevent the activation of ATF4 following

OPA1 knockout (Figure 4N). We also note that these data were included in the original submission.

Reviewer #2: Thank you for your helpful remarks and suggestions, as we feel these have substantially improved the manuscript. Please see below for answers to each question you raised in your initial review of the manuscript.

Q1: Most notably, an over-reliance on Mdivi-1, a drug with well known off target effects on ETC function, makes it difficult to interpret much of the pharmacological data.

A1: We thank the reviewer for pointing this out and ask that the reviewer please see our response to Reviewer #1 Q7 (first two paragraphs).

Q2: There is a failure to consider alternative and quite plausible mechanisms through which *Opa1* inhibition might be affecting apoptosis sensitivity.

A2: We thank the reviewer for the comments related to the *OPA1* mechanism and ask that the reviewer see our answers to his or her specific points in the answers below.

Q3: In the abstract the authors claim: "...there have been no studies to define therapeutic vulnerabilities resulting from these alterations." and in the intro they claim: "Despite the observation that mitochondrial dynamics are frequently altered in human cancers and the likelihood that these alterations broadly impact cell physiology, there have been no efforts to define therapeutic vulnerabilities driven by altered dynamics." - This is an interesting study and their approach has potential to provide valuable insights, but this claim seems to be an unnecessary attempt to claim novelty.

A3: We agree with the reviewer that we did not need to include such statements in the manuscript. We have since removed them.

Q4: Most notably, several of the genes analyzed in this figure reside in genomic regions with other genes whose amplification can explain the differences in drug sensitivity as well or better than the gene being highlighted. For example, *MFN1*, the most highly amplified gene in Figure 1A, lies in a region of chromosome 3 within 150 kilobases of *PIK3CA*. Similarly, *DNM1L* is near the *KRAS* gene on chromosome 12, the amplification of which is validated to play a role in cancer and may better explain the differential sensitivity to PI3K pathway inhibitors. To draw conclusions about the potential relationship between mitochondrial dynamics machinery amplification and drug sensitivity from this data, the authors would need to control for amplification of nearby oncogenes in each of these regions to see if the correlations that have been identified are robust.

A4: This is an excellent point and we have re-done the analyses by controlling for neighboring oncogenes where appropriate: *KRAS* (*DNM1L*), *BRAF* (*FIS1*), *PIK3CA* (*MFN1* and *OPA1*), and *mTOR* (*MFN2*). The exclusion of cell lines in which both dynamics-regulating genes and a neighboring oncogene were amplified significantly limited our power to detect drug sensitivity differences for several of the genes. However, in the cases where we were powered to perform the appropriate analyses, we were actually able to detect more altered sensitivities than in the previous analysis. For example, *DNM1L* amplifications are

now associated with close to 35 drug sensitivity differences after excluding cell lines with co-amplification of the nearby oncogene (Figure 1B). In addition, we were able to detect an XIAP inhibitor, embelin, as being specifically potent in *DNM1L* amplified breast cancer cells—consistent with the work conducted in the rest of the manuscript (Figure 1C,D). Lastly, all PI3K pathway inhibitors that previously scored in *DNM1L* amplified breast cancers remained in our updated analysis after we controlled for concurrent *KRAS* amplifications (Figure 1C,D).

Q5: It has long been appreciated that Mdivi-1 has numerous off target effects and a recent study demonstrates both that it is a Complex I inhibitor and potentially a poor Drp1 inhibitor (Bordt et al Dev Cell. 2017). For this reason, the genetic approaches presented in this figure are much more compelling than the pharmacological approaches. Complex I inhibition by Mdivi-1 will have a lot of downstream effects (see the literature on metformin) so unfortunately, this drug is not a very useful tool for understanding the biology of mitochondrial fission.

A5: Again, we thank the reviewer for this comment and ask that he or she refer to our response to Reviewer #1 Q7 (first two paragraphs).

Q6: The western blots in 4B need some controls (mitochondrial proteins to show purity of cytoplasmic fraction, e.g.).

A6: We thank the reviewer for pointing this out. We have now included AIF as a control for this experiment in Figure 4B.

Q7: Also, the western data as presented (both cyt. C and caspase 9) is consistent with incomplete MOMP (iMOMP) in a large fraction of the cells or complete MOMP in a small fraction of the cells. I.F. would be able to distinguish these possibilities, the former of which would be consistent with the cyt. C leakiness proposed by the authors.

A7: This is an excellent suggestion by the reviewer. We have performed cytochrome c IF to try to distinguish this possibility. We see that in the case of *DNM1L* knockout, on average, co-localization (Pearson correlation of two image channels) of cytochrome c to the mitochondria is decreased when compared to control knockout cells (Supplemental Figure 4G,H). Further, we see no evidence of increased numbers of cells with complete MOMP relative to control cells. Together, these data are consistent with incomplete leakiness of cytochrome c in a large fraction of cells rather than a small fraction of the population undergoing complete MOMP. We have noted this issue in the revised text.

Q8: The data on Opa1 are compelling, but the model ignores an additional aspect of Opa1 biology that might explain the data. Opa1 is important for maintenance of cristae junctions and its cleavage is important for complete cytochrome c release, as much of the cytochrome c can otherwise be trapped in the cristae folds. It is possible that Opa1 deletion is allowing for cytochrome c release (but maybe not IAP release) from mitochondria under conditions where it is normally prevented (ie - Bax/Bak independent damage to OMM). This would also explain the sensitivity to SMAC mimetics and is consistent with a known role for Opa1.

A8: This is an excellent point, and in fact we, too, thought that Opa1 loss could lead to leakiness of cytochrome c, especially given the biological observations described by the reviewer. However, despite substantial effort, we were unable to observe this effect in OPA1 knockout cells, as indicated by the representative blots, taken from two distinct cell lines, in Supplementary Figure 4I and 4J. In fact, it was this very observation that compelled us to look into alternative mechanistic explanations, work which ultimately led to the UPR findings described in the manuscript.

Q9: The authors consistently write that Opa1 inhibition increases mitochondrial fission and Drp1 inhibition increases fusion. They should be more careful with the language here and use more precise wording such as “fragmentation” and “connectivity”. Fusion activity is not directly affected by inhibition of fission nor is fission activity affected by fusion inhibition.

A9: We thank the reviewer for drawing this important distinction. We have edited the manuscript and figures to reflect this point, using words like “connectivity” and “fragmentation” instead of “fission” or “fusion”.

Q10: For images of mitochondrial morphology, the authors show limited replicates and perform no quantitation of the observed phenotypes.

A10: We have now quantified all images using an established approach (Wang et al. 2017). We plot mitochondrial length/width of thousands of mitochondria across $n > 10$ cells from at least two independent experiments (typically three independent experiments).

Q11: It is never examined whether there is any relationship between the copy number alterations and mitochondrial morphology, even for a small subset of available cell lines. This would strengthen the arguments made in the paper, especially for figure 1 and 3D

A11: This is an excellent suggestion. Although an extensive analysis of the relationship between copy number alterations and mitochondrial morphology is beyond the scope of this work, we have nevertheless performed preliminary studies to address this question. In breast cancer cell lines lacking or harboring amplifications (i.e. AU565, which lack amplifications, and MDA-MB-436, which harbor *DNM1L* and *FIS1* amplifications) we observed that AU565 mitochondrial morphology is more connected than MDA-MB-436 (Supplemental Figure 1A). In addition, in a comparison of the melanoma cell lines SKMEL2 (*OPA1* amplified) and UACC62 (wild-type), SKMEL2 were observed to possess a more connected mitochondrial network compared to UACC62 (Supplemental Figure 1B). However, in a comparison of the *KRAS* mutant pancreatic cell lines ASPC-1 (wild-type) and Panc02.03 (*OPA1* amplified), we were unable to detect a difference in the connected/fragmented morphology of the mitochondrial network (Supplemental Figure 1C). This is perhaps due to the well-established fact that *KRAS* mutations lead to the constitutive activation of the ERK pathway, ultimately resulting in Drp1 activation (Kashatus et al. 2015; Serasinghe et al. 2015). Together, these data begin to suggest that amplifications in dynamics regulating genes may affect mitochondrial morphology in predictable ways, inviting future studies to systematically dissect the intricacies of the relationship between amplifications and mitochondrial morphology.

REVIEWERS' COMMENTS:

Reviewer #2 (Remarks to the Author):

The authors have addressed a number of my concerns, and I find the manuscript improved. However, there remain aspects of this study with which I am struggling. In particular, in the response "A-11" the authors state, "extensive analysis of the relationship between copy number alterations and mitochondrial morphology is beyond the scope of this work". Yet this is really a key underlying assumption of the whole study. For the authors to state, as they do in the abstract, that "alterations in the structural dynamics of this organelle result in unique therapeutic vulnerabilities" they need to characterize more than alterations in the machinery or change the way they describe their conclusions. Throughout the manuscript they link changes in the levels of the fusion/fission machinery (either DNA, RNA or Protein) to changes in mitochondrial dynamics, and conclude that the phenotypic changes they observe are the result of those dynamics changes. They really need to be more careful with language and to not over-conclude from their limited data.

In terms of mechanism, while the authors have added several controls, as requested, to the experiments in figure 4 and the related supplement, I remain uncomfortable with the broad conclusions being drawn from the limited level of analysis presented. Alternative mechanisms for how changing mitochondrial shape can change apoptosis sensitivity have been described in the literature (For example, Renault, et al Mol. Cell 2015), with much more supporting data, yet these mechanisms are not addressed and the studies not cited. I think to propose a model, or to draw broad conclusions, as the authors do, requires a stronger case to be made and alternative mechanisms to be acknowledged.

Response to Reviewers' comments:

Reviewer #1: Thank you for the time you took to review our manuscript.

Reviewer #2: Thank you for your helpful remarks, as we feel these have substantially improved the manuscript. Please see below for responses to each point you raised in your re-review of the manuscript.

1: In particular, in the response "A-11" the authors state, "extensive analysis of the relationship between copy number alterations and mitochondrial morphology is beyond the scope of this work". Yet this is really a key underlying assumption of the whole study. For the authors to state, as they do in the abstract, that "alterations in the structural dynamics of this organelle result in unique therapeutic vulnerabilities" they need to characterize more than alterations in the machinery or change the way they describe their conclusions.

1: We agree that our study does not do a comprehensive analysis of the relationship between copy number alterations and mitochondrial morphology. However, we find that this point is only one aspect of our study and we provide starting evidence that such a relationship exists. Further, through isogenic cell line models, we are able to demonstrate the connection between mitochondrial dynamics protein perturbations and drug sensitivity. To address your concerns, we have altered our wording throughout the manuscript as it pertains to mitochondrial structure and copy number amplifications--making sure to highlight the premature nature of this relationship for the reader.

2: Throughout the manuscript they link changes in the levels of the fusion/fission machinery (either DNA, RNA or Protein) to changes in mitochondrial dynamics, and conclude that the phenotypic changes they observe are the result of those dynamics changes. They really need to be more careful with language and to not over-conclude from their limited data.

2: We have altered the wording throughout to text to make sure that we talk specifically about changes to either DNA, RNA, or protein as being related to our phenotype, rather than concluding that this is through dynamics changes.

3: Alternative mechanisms for how changing mitochondrial shape can change apoptosis sensitivity have been described in the literature (For example, Renault, et al Mol. Cell 2015), with much more supporting data, yet these mechanisms are not addressed and the studies not cited. I think to propose a model, or to draw broad conclusions, as the authors do, requires a stronger case to be made and alternative mechanisms to be acknowledged.

3: This is a great point and we agree that additional mechanisms could be working in tandem with our mechanisms to explain the sensitivity to SMAC mimetics. We have highlighted this for the reader and updated our citations to include Renault et al. Mol. Cell 2015.

We thank you again for considering our work.